# ErrorTrace: A Black-Box Traceability Mechanism Based on Model Family Error Space

Chuanchao Zang[1], Xiangtao Meng[1], Wenyu Chen[1], Tianshuo Cong[2,6], Yaxing Zha[3], Dong Qi[4],
Zheng Li[1,5,6,*], Shanqing Guo[1,5,6,*]

[1]*School of Cyber Science and Technology, Shandong University*
[2]*School of Cryptologic Science and Engineering, Shandong University*
[3]*China Transportation Information Technology Group Co., Ltd., China*
[4]*Shandong Branch, China United Network Communications Group Co., Ltd., China*
[5]*State Key Laboratory of Cryptography and Digital Economy Security, Shandong University*
[6]*Shandong Key Laboratory of Artificial Intelligence Security, Shandong University*

## Abstract

The open-source release of large language models (LLMs) enables malicious users to create unauthorized derivative models at low cost, posing significant threats to intellectual property (IP) and market stability. Existing IP protection methods either require access to model parameters or are vulnerable to fine-tuning attacks. To fill this gap, we propose `ErrorTrace`, a robust and black-box traceability mechanism for protecting LLM IP. Specifically, `ErrorTrace` leverages the unique error patterns of model families by mapping and analyzing their distinct error spaces, enabling robust and efficient IP protection without relying on internal parameters or specific query responses. Experimental results show that `ErrorTrace` achieves a traceability accuracy of *0.8518* for *27* base models when the suspect model is not included in `ErrorTrace`'s training set, outperforming the baseline by *0.2593*. Additionally, `ErrorTrace` successfully tracks *34* fine-tuned, pruned, and merged models across various scenarios, demonstrating its broad applicability and robustness. In addition, `ErrorTrace` shows a certain level of resilience when subjected to adversarial attacks. Our code is available at: https://github.com/csdatazcc/ErrorTrace.

## 1 Introduction

Large language models (LLMs) have advanced rapidly in recent years, with increasing model sizes and expanding training datasets significantly improving their capabilities in language understanding and generation (1; 20; 28; 47). However, these advancements come at the cost of requiring both substantial computational resources and deep domain expertise (27), raising critical concerns about model copyright and intellectual property (IP) rights. One major concern is the risk of unauthorized use and repackaging of original models. Malicious actors can fine-tune open models and claim them as their own, undermining the contributions of the original developers (10; 26; 27; 28). A notable case in 2024 involved a group of Stanford University students allegedly plagiarizing MiniCPM-Llama3-V2.5, a Chinese open-source model developed by Facade Intelligence (39).[1]

To protect LLM IP, various traceability methods have been proposed. Watermarking, the most prominent approach, typically requires modifying the original model parameters (2; 3; 15; 16; 25; 34; 38; 43), while other methods avoid modifying the model but still require access to its parameters (21; 42; 37; 40; 41). Thus, these methods are limited to white-box settings, which restricts their applicability. Black-box query-based methods rely on carefully designed prompts but often

---

[1]Corresponding authors

[1]https://www.globaltimes.cn/page/202406/1313632.shtml

involve high computational costs or require human expertise. They are also sensitive to changes in model behavior, such as those introduced by fine-tuning, which limits their robustness (8; 11; 18).

To fill this gap, we propose implicit fingerprinting via data-driven error analysis—a black-box mechanism that identifies the likely family origin of a suspicious model based solely on its error patterns. Instead of relying on white-box access or carefully crafted queries, our method analyzes the statistical consistency of prediction errors to uncover distinctive failure behaviors shared within a model family. The core insight driving our method is that *how models fail* can be more revealing than where they succeed. While correct outputs may arise from convergent behaviors across independently trained models, consistent and uncommon errors often reflect deeper shared characteristics—such as architectural biases, training regimes, or optimization strategies (see details in Appendix D). Analogous to human behavior, rare mistakes shared by multiple individuals can imply similar training or background; likewise, models from the same family tend to exhibit stable and structured error patterns, whereas models from different families display more diverse failure distributions. This phenomenon is empirically validated in Figure 1a, which shows significantly higher error similarity within families than across unrelated models (see section 3 for details).

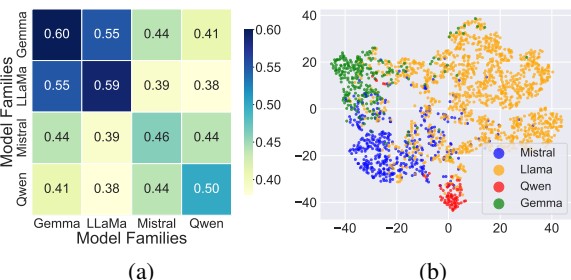

Figure 1: (a) Intersection of erroneous data from different family models. (b) Error space dimensionality reduction display.

Building on this insight, we introduce `ErrorTrace`, a black-box traceability framework for LLM IP protection. Given a set of data samples, we extract each model family's error patterns and embed them into an error space via graph-based analysis (illustrated in Figure 1b). A suspect model's error behavior is then projected into this space to infer its likely family affiliation, enabling reliable lineage attribution without internal access or watermarking.

We evaluate our method on five benchmark datasets and 27 models from four popular LLM families. The results show that our method achieves 100% traceability accuracy when the suspect model exists in `ErrorTrace`'s model set, and 85.18% when it does not exist, outperforming the baseline by 25.93%. Additionally, our method performs well across various fine-tuning, pruning, and merging strategies. In addition, `ErrorTrace` shows a certain level of resilience when subjected to adversarial attacks.

**Contributions**. Our contributions are three-fold:

- A novel traceability perspective: We introduce a new approach to model traceability by analyzing error space differences across model families, providing new insights into LLM IP protection.

- An effective and adaptable method: We construct a model family error space using unique error patterns and graph-based connections, enabling reliable traceability even among models with similar architectures.

- Comprehensive evaluation: We validate our method on 27 models from four families, which range from 3B to 141B, this marks the first instance of achieving such a wide range of traceability verification. Our approach outperforms baselines and accurately traces all fine-tuned, pruned, and merged models, demonstrating its effectiveness and robustness.

## 2 Threat Model and Related Works

### 2.1 Threat Model

The auditor, either the original model owner or a third-party detection agency, has full access to the original model but only black-box access to the suspect model, i.e., <query, output> pairs.

A suspect model is one modified through unauthorized fine-tuning or other actions that infringe on the original model's IP. It typically has two characteristics: similarity to the original model in architecture and knowledge, and concealment of its link to the original model.

## 2.2 Traceability Methods

In the LLM era, traceability methods can be broadly classified into three categories.

**Modifying Model Parameters**: Researchers have explored embedding watermarks into LLMs (2; 3; 12; 13; 15; 16; 19; 23; 25; 31; 34; 35; 36; 38; 43; 46). However, these methods require model modifications, increasing training costs and potentially impacting performance. Besides, it cannot be applied to released models without protection unless retrained, limiting its practicality.

**Accessing Model Parameters**: These methods compare internal parameters, features, or output logits to assess model similarity (21; 37; 40; 41; 42). However, their practicality is limited since most major LLMs provide only external APIs.

**Query-based Traceability**: These methods identify suspect models by analyzing responses to specific queries. TRAP (8) ,ProFLingo (11) generate a query and expect a specified response, but are limited by high query costs and variable response rates in fine-tuned models. LLMmap (18) assesses output similarity using empirical queries, but its robustness declines against adversarial obfuscation.

## 2.3 Traceability Granularity

Model-level traceability is widely used in LLM IP protection to identify the specific base model behind a suspect. However, existing black-box methods often lack robustness and are vulnerable to interference (38). For large-scale developers (e.g., Meta), protecting every individual model is computationally costly. In contrast, model family-level traceability focuses on identifying the broader model family, reducing overhead while aligning with the needs of developers—key stakeholders in IP disputes. By leveraging shared features within a family, it remains effective even after fine-tuning or pruning. It also avoids dependence on fixed queries, offering greater robustness against adaptive attacks and input perturbations.

# 3 Exploring the Potential of Error Space for Traceability

This section explores the feasibility of using model error spaces for traceability. We present a probabilistic model showing that error patterns are statistically distinct across model families and validate this through experimental simulations.

Consider model families $G_1, G_2, \ldots, G_n$, each containing models $M_1, M_2, \ldots, M_m$, and a dataset $D = \{d_1, d_2, \ldots, d_N\}$ with $N$ samples. For model $M_i$ and data $d_j$, define the binary random variable $E_{ij}$ as follows:

$$E_{ij} = \{ \begin{array}{ll} 1 & \text{if } M_i \text{ predicts incorrectly on } d_j \\ 0 & \text{otherwise} \end{array} \tag{1}$$

Each model's error pattern is represented by the binary vector $\mathbf{E}_i = (E_{i1}, \ldots, E_{iN})$. Model traceability aims to identify the family $G_k$ based on $\mathbf{E}_i$. For mathematical validation, we make the following assumptions:

1. Independence Assumption: The error events on different data points are independent.

2. Family Distinguishability Assumption: Models from different families have significantly different error probabilities on certain data points.

3. Completeness Assumption: The error events for all models on all data points are observable and recorded completely.

| Number | Rate | LR | DT | RF |
|---|---|---|---|---|
| | 0.01 | 0.00 | 0.60 | 0.20 |
| 5000 | 0.05 | 0.60 | 0.40 | 0.80 |
| | 0.1 | 0.60 | 0.20 | 0.40 |
| | 0.01 | 0.60 | 0.20 | 0.60 |
| 10000 | 0.05 | 1.00 | 0.40 | 0.80 |
| | 0.1 | 0.60 | 0.40 | 0.60 |
| | 0.01 | 0.80 | 0.00 | 0.20 |
| 50000 | 0.05 | 1.00 | 0.20 | 0.60 |
| | 0.1 | 1.00 | 0.20 | 0.60 |
| **Average** | | **0.69** | 0.29 | **0.53** |

Table 1: The tracing success rate for logistic regression (LR), decision trees (DT), and random forests (RF) is evaluated across varying data quantities and proportions of erroneous data.

Consider the suspicious model $M_i$ and its error pattern $\mathbf{E}_i$ on dataset $D$. Assuming independence, the error event $E_{ij}$ for each data point $d_j$ is independent. Thus, the joint probability distribution of $\mathbf{E}_i$ is the product of the individual error event probabilities. To determine

the model family of $M_i$, the problem reduces to comparing the joint probability distribution of $\mathbf{E}_i$ across different model families.

$$P(\mathbf{E}_i|M_i \in G_k) = \prod_{j=1}^{N} P_{G_k,j}^{E_{ij}}(1 - P_{G_k,j})^{(1-E_{ij})} \tag{2}$$

where $Pr_{G_k,j} = P(E_{ij} = 1|M_i \in G_k)$ is the error probability of family $G_k$ on data point $d_j$.

Based on the assumption of family distinguishability, for model families $G_a$ and $G_b$, there exists a data point $j$ such that $P_{G_a,j} \neq P_{G_b,j}$. Thus, the joint probability distribution of the error pattern $\mathbf{E}_i$ differs across model families. The suspicious model can be assigned to a family based on these probabilistic differences.

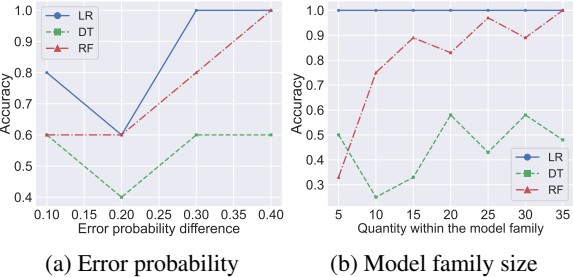

(a) Error probability      (b) Model family size

Figure 2: (a) The impact of error probability difference on traceability accuracy; (b) The impact of model family size on traceability accuracy.

To validate our approach, we run a traceability experiment using a synthetic dataset across four model families. Each family has defined error-prone data points with varying error probabilities. We use Logistic Regression (LR), Decision Trees (DT), and Random Forests (RF) for classification. As shown in Table 1, each family includes four models with a 0.2 error rate difference. LR and RF both achieve accuracy above 0.5, supporting the feasibility of using error patterns for traceability.

We further analyze the effect of model count and error rates using 10,000 data points at a 0.1 error rate. Figure 2 shows LR and RF improve with more models and greater error differences, while DT is less consistent. This indicates that more erroneous data enhances traceability. (See Appendix K for details.)

## 4 Methodology of `ErrorTrace`

We propose `ErrorTrace`, a novel black-box traceability method for LLM IP protection. As shown in Figure 3, `ErrorTrace` consists of three steps: error uniqueness calculation, error space construction, and suspect model inference. This section explains how the classification benchmark constructs the error space; the regression case is provided in Appendix B.

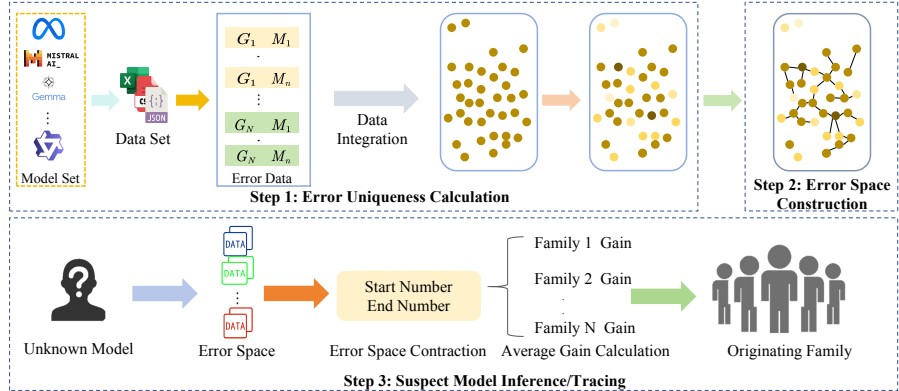

Figure 3: The overview of error space-based traceability mechanism `ErrorTrace`.

### 4.1 Error Uniqueness Calculation

For a model family $T$ with $N$ LLMs from the same familis, we first sample a prompt dataset $S$ covering various topics to approximate the input space. Each prompt in $S$ represents a data point $s$. We then calculate the error uniqueness of each data point $s$ for the model family $T$, considering two key aspects:

**Intra-family Error Uniqueness.** This aspect calculates the error uniqueness of data point $s$ across all models in family $T$, represented by $U$. The simplest approach is to calculate the proportion of models in $T$ that make errors on $s$. However, we observe that higher-performing models better reflect the family's unique error characteristics; thus, we weight errors by model accuracy, assigning greater weight to errors from high-performing models, as shown in Equation 3.

$$U = \sum_{j \in E} p_j \bigg/ \sum_{i=1}^{N} p_i \tag{3}$$

Here, $E$ denotes the set of LLMs in the target family that generate an error response on data point $s$, and $p$ represents the accuracy of the test model on the topic related to $s$.

**Cross-family Error Discrepancy.** This aspect verifies whether the error patterns at data point $s$ are unique to model family $T$. We introduce the cross-family error discrepancy $PF$ as a penalty factor to filter out shared error data points across different families. Instead of treating all non-target families as a group, we compute the error rate for each non-target family at $s$ and assign higher penalty weights to families with higher error rates using squared summation, as shown in Equation 4.

$$PF = 1 - \sum_{j \in G} P_j^2 \bigg/ \sum_{i=1}^{K} P_i \tag{4}$$

Let $G$ represent the set of model families whose models produce errors on data point $s$, $P_j$ denote the error rates of $s$ for family $j$, and $K$ be the number of model families excluding family $T$.

To better mitigate the impact of error-prone points in non-target families, we incorporate model performance into the calculation of $PF$. The error rate $P$ for each non-target family at data point $s$ is defined as:

$$P_j = \sum_{j \in E} p_j \bigg/ \sum_{i=1}^{N} p_i \tag{5}$$

Thus, the error uniqueness of data points $s$ in a certain model family $T$ can be expressed as:

$$WP_{(s,T)} = U \times PF \tag{6}$$

## 4.2 Error Space Construction

In the previous section, we calculated the error uniqueness for all data points in the model family $T$. Now, we construct the error space for $T$ using a graph-based approach. Unlike traditional methods like K-Means (17), DBSCAN (5), and GMM (22), we avoid this limitation by assigning each data point a weight based on its error uniqueness. Edge weights between data points are determined by their Euclidean distances, which we normalize using the Cumulative Distribution Function (CDF) (6) to reduce the impact of extreme values.

$$F(D_{(i,j)}) = P(D \leq D_{(i,j)}) \tag{7}$$

Where $D(i,j)$ represents the distance between data points $i, j$ and F is the CDF function.

Using a single CDF mapping overlooks the effect of adjacent distance intervals. To address this, we introduce an interval-based CDF weighting mechanism. We first sort all edge distances $D_{(i,j)}$ in ascending order to obtain a sequence $d_1, d_2, \ldots, d_n$, where $n$ is the total number of edges and $d_i$ is the $i^{\text{th}}$ distance. Then define $w_j$ as the difference between consecutive sorted distances $d_{j+1}$ and $d_j$. The new mapping formula is defined as:

$$WE(d_i) = \sum_{j=1}^{i-1} w_j \bigg/ \sum_{k=1}^{n-1} w_k \tag{8}$$

where $WE(d_i)$ represents the weighted normalized value of the distance $d_i$.

To improve computational efficiency, we use a grid search approach to identify the densest region of edge weights (denoted as $dense$), and the filtered edge set $L$ is defined as:

$$L = Edge(WE(d_i) \leq dense) \qquad (9)$$

where $Edge$ represents the original set of edges, and $L$ is the filtered set of edges.

We construct candidate error spaces for each model family $T$ using graph-based connections. Starting from the point with the highest error uniqueness, we apply BFS (4) to connect neighboring points. A point is added if its uniqueness exceeds the edge weight; otherwise, it is skipped. This process continues until no further connections can be made. We repeat it for remaining unassigned points to form all candidate error spaces (see Figure 4).

To determine the optimal error space for model family $T$, we compute the effective uniqueness sum for each dataset and select the error space with the highest sum as the optimal error space. This is denoted as:

$$R(k) = \sum_{s \in S} WP(s) - \sum_{l \in L'} WE(l) \qquad (10)$$

where $k$ is the set of candidate error spaces, $S$ represents all points contained in the space, $L'$ represents all edges used in the connection.

### 4.3 Suspect Model Inference

For each error space, we progressively reduce its size by increasing the minimum error uniqueness. For a suspect model $M_s$, we track the average error rate change during this contraction. We compute $\Delta$, the difference between the maximum change and the average of the remaining changes. If $\Delta < \tau$, the model shows no strong alignment with any family and is considered out-of-distribution. If $\Delta \geq \tau$, the family with the largest change is identified as the source. To ensure fairness, we normalize contraction

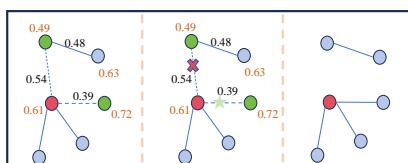

Figure 4: Example diagram of error space construction

by fixing the number of remaining points at the start and end. See details in Appendix C.

## 5 Experiment

### 5.1 Experimental Setup

**Models**: We use 27 models to build error space, ranging from 3 B to 141 B parameters, across families like LLaMa, Qwen, Mistral, and Gemma (see Appendix A).

**Datasets**: We use five benchmark datasets namely Multinli (33), PAWS (45), CoLA (32), AG News (44), and CommonsenseQA (24) to find error data. See details in Appendix N.

**Baselines**: Since `ErrorTrace` is a black-box traceability mechanism, we compare with black-box baselines, i.e., LLMmap (18), TRAP (8), and ProFLingo (11), which use query-based tracing. Due to space limitation, we introduce TRAP (8), ProFlingo (11) and the LLMmap (18) processing in Appendix F.

**Metrics**: We evaluate traceability using three metrics: accuracy (ACC), cosine distance, and model family discrepancy (MFD). Cosine distance measures the similarity between a test model's features and the mean features of each family—a smaller value indicates stronger alignment. This also allows direct comparison with LLMmap, which uses a similar approach. MFD detects out-of-distribution (OOD) models by measuring how much a test model's embedding deviates from the structure of known model families.

**Traceability Scenario**: We consider three scenarios to evaluate `ErrorTrace`, namely *model set known*, *model set unknown*, and *real-world scenario*. In the model set known scenario, all base models contribute to constructing the error space, while also serving as test models. In model set unknown scenario, we designate *one* model as the suspect and construct error spaces using the

| | Model | LLMmap | | | | | Ours | | | | |
|---|---|---|---|---|---|---|---|---|---|---|---|
| | | **Gemma** | **Qwen** | **LLaMa** | **Mistral** | **ACC** | **Gemma** | **Qwen** | **LLaMa** | **Mistral** | **ACC** |
| **LLaMa** | LLama7b | **0.327** | 0.376 | 0.386 | 0.561 | ✗ | 1.188 | 1.293 | **0.009** | 1.012 | ✓ |
| | LLama70b | 0.341 | 0.432 | **0.303** | 0.599 | ✓ | 1.166 | 1.346 | **0.007** | 1.01 | ✓ |
| | LLama3:8b | 0.405 | 0.445 | **0.311** | 0.562 | ✗ | 1.370 | 1.227 | **0.360** | 1.660 | ✓ |
| | LLama3:70b | 0.388 | 0.489 | **0.226** | 0.526 | ✓ | 1.643 | 1.170 | **0.283** | 1.304 | ✓ |
| | LLama3.2:3b | **0.366** | 0.426 | 0.398 | 0.506 | ✗ | 1.105 | 1.241 | **0.039** | 1.387 | ✓ |
| | LLama3.1:8b | **0.366** | 0.509 | 0.433 | 0.564 | ✗ | 0.928 | 1.716 | **0.218** | 0.581 | ✓ |
| | LLama:13b | **0.327** | 0.376 | 0.386 | 0.561 | ✗ | 1.802 | 0.689 | **0.492** | 0.758 | ✓ |
| **Gemma** | Gemma7b | 0.606 | 0.574 | **0.469** | 0.560 | ✗ | **0.641** | 1.667 | 1.324 | 1.116 | ✗ |
| | Gemma2:9b | 0.333 | 0.363 | **0.268** | 0.568 | ✗ | 1.662 | **0.299** | 1.313 | 0.567 | ✗ |
| | Gemma2:27b | **0.217** | 0.390 | 0.426 | 0.579 | ✓ | 1.928 | **0.238** | 1.065 | 1.087 | ✗ |
| **Qwen** | Qwen7b | 0.315 | **0.110** | 0.375 | 0.519 | ✓ | 1.237 | **0.120** | 1.116 | 0.970 | ✓ |
| | Qwen72b | 0.342 | **0.068** | 0.468 | 0.577 | ✓ | 1.382 | **0.013** | 1.431 | 1.317 | ✓ |
| | Qwen3b | 0.296 | **0.118** | 0.471 | 0.563 | ✓ | 1.270 | **0.096** | 1.007 | 1.321 | ✓ |
| | Qwen32b | 0.423 | **0.229** | 0.459 | 0.538 | ✓ | 1.597 | **0.018** | 1.381 | 1.233 | ✓ |
| | Qwen2:7b | 0.258 | **0.157** | 0.386 | 0.507 | ✓ | 1.236 | **0.058** | 1.521 | 1.372 | ✓ |
| | Qwen14b | 0.295 | 0.304 | **0.293** | 0.504 | ✗ | 1.466 | **0.016** | 1.366 | 1.397 | ✓ |
| | Qwen1.5:7b | **0.246** | 0.385 | 0.403 | 0.496 | ✗ | 1.355 | **0.399** | 0.625 | 0.814 | ✓ |
| | Qwen1.5:72b | 0.585 | **0.469** | 0.506 | 0.601 | ✓ | 1.590 | **0.046** | 1.076 | 1.407 | ✓ |
| | Qwen1.5:4b | 0.429 | **0.156** | 0.403 | 0.504 | ✓ | 1.412 | **0.003** | 1.363 | 1.273 | ✓ |
| | Qwen1.5:32b | 0.406 | **0.262** | 0.281 | 0.493 | ✓ | 1.024 | **0.136** | 1.482 | 0.987 | ✓ |
| | Qwen1.5:14b | 0.474 | 0.482 | 0.539 | **0.117** | ✗ | 1.487 | **0.038** | 1.240 | 1.001 | ✓ |
| **Mistral** | Mixtral | 0.434 | 0.487 | 0.483 | **0.142** | ✓ | 0.893 | 0.577 | 1.747 | **0.401** | ✓ |
| | Mistral7b | 0.370 | 0.471 | **0.343** | 0.617 | ✗ | 0.710 | 0.679 | 1.564 | **0.335** | ✓ |
| | Mistral:small | 0.457 | 0.491 | 0.483 | **0.085** | ✓ | 1.026 | 0.930 | 1.335 | **0.055** | ✓ |
| | Mistral:nemo | 0.543 | 0.521 | 0.537 | **0.057** | ✓ | 1.669 | 1.150 | 0.821 | **0.422** | ✓ |
| | Mistral:large | 0.588 | 0.508 | 0.496 | **0.207** | ✗ | 1.1241 | **0.137** | 1.172 | 1.303 | ✗ |
| | Mixtral:8x22B | 0.306 | **0.237** | 0.275 | 0.351 | ✗ | 1.389 | **0.293** | 1.818 | 0.819 | ✗ |
| | | | | | | **0.59** | | | | | **0.85** |

Table 2: Cosine similarity in model set unknown scenario

remaining *26* models, this is a more challenging scenario. In the real-world scenario, the suspect model is modified through a series of fine-tuning, pruning, and merging methods.

**Prompt Template**: To minimize information leakage, we restrict prompt wording, limiting the suspect model's association with its family. All prompt templates are provided in Appendix L.

## 5.2 Effectiveness

This section evaluates our method in three key areas: error space construction, traceability of models, and comparison of space search methods. We define the error space contraction process, setting start and end numbers at 1200 and 600 points (the reasons for selection are shown in Figure 6b and Appendix E).

First, our method effectively captures each model's error space, clearly separating different families. As shown in Figure 5a, error data across families have near-zero overlap, confirming the isolation of family-specific patterns. In the known model set scenario, all 27 test models are correctly traced to their families. Figure 5b further shows that each family has positive alignment within its own space and negative alignment elsewhere. See Appendix I for details.

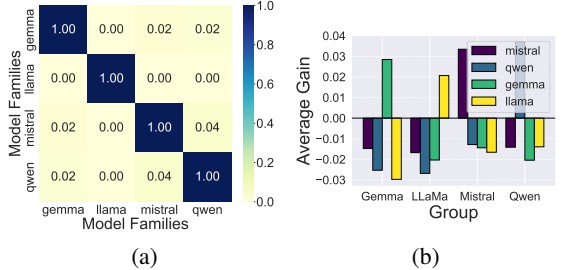

Figure 5: (a) Model family error space error intersection; (b) Average change on different error spaces.

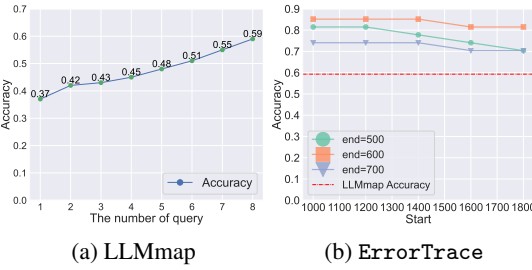

(a) LLMmap  (b) `ErrorTrace`

Figure 6: (a) The effect of query numbers on LLMmap; (b) Comparison of the highest accuracy of `ErrorTrace` and LLMmap.

Second, our method performs well in the more challenging unknown model set scenario. Each model is tested as a suspect using the remaining 26 to build error spaces. As shown in Table 2, our method achieves 85.18% accuracy (23/27), significantly outperforming LLMmap's 59.25% (16/27).

In this scenario, four models were misidentified. Two were from the Gemma family, likely due to having only two models left to construct a reliable error space. LLMmap also struggled with these. The other two were large Mistral models (Mixtral:8x22B and Mistral-large), whose scale distribution is uneven (see details in section 7).

Furthermore, our method outperforms traditional space construction techniques (e.g., K-Means, DBSCAN, GMM) in identifying effective error spaces for model traceability. Using grid search for fair comparison, the

| Methods | K-Means | DBSCAN | GMM | Ours |
|---------|---------|--------|-----|------|
| **Accuracy** | 0.4074 | 0.4444 | 0.6666 | **0.8518** |

Table 3: Accuracy of different construction methods.

best baseline achieves only 66.66% accuracy, well below our 85.18% (Table 3). We also evaluate robustness to contraction intervals. Figure 6a shows LLMmap peaks at 59.25% when using all queries. In contrast, our method maintains high accuracy across varying intervals and consistently outperforms LLMmap (Figure 6b). These results underscore the superiority of our approach in error space identification and model traceability.

## 5.3 Impact of Dataset Size

We also test how dataset size affects `ErrorTrace` by randomly sampling from the original error set (Table 4). Performance improves with more data, and even at 20K samples, `ErrorTrace` significantly exceeds LLMmap's best result.

| Size | 20K | 30K | 40K | 50K |
|------|-----|-----|-----|-----|
| **Accuracy** | 0.7037 | 0.7778 | 0.8148 | **0.8518** |

Table 4: Accuracy at different dataset sizes.

## 5.4 Out-Of-Distribution Model Detection

We designate LLaMA as the out-of-distribution family and treat Gemma, Qwen, and Mistral as in-distribution. Using an optimal threshold $\tau = 0.025$, we compute model family discrepancy for each model. For out-of-distribution evaluation, we test three unseen models: Falcon3-7B and Falcon3-10B (different scales within the Falcon family), and DeepSeek-R1-7B (a different family). Table 5 shows that all these models have discrepancies below $\tau$, confirming they are correctly identified as out-of-distribution and not part of any known family. This indicates that we can correctly identify these out-of-distribution models as not belonging to any of the in-distribution families.

| Model | Gemma | Qwen | LLaMa | Mistral | MFD | ACC |
|-------|-------|------|-------|---------|-----|-----|
| Falcon3-7B | 0.0014 | 0.0019 | 0.0014 | 0.0017 | 0.0124 | ✓ |
| Falcon3-10B | -0.0112 | -0.0099 | -0.0189 | -0.0164 | 0.0167 | ✓ |
| DeepSeek-R1-7B | 0.0067 | 0.0073 | -0.0011 | 0.0012 | 0.0151 | ✓ |

Table 5: Out-of-distribution model detection.

1. To visualize the MDF, this table presents the average change in the suspect model's error rate within each model family's error space, rather than cosine distance.

## 5.5 Robustness

| Method | Model | Gemma | Qwen | LLama | Mistral | ACC |
|--------|-------|-------|------|-------|---------|-----|
| **LLMmap** | mistral-openorca | 0.360 | 0.347 | **0.308** | 0.413 | ✗ |
| | mistrallite | 0.607 | 0.609 | **0.593** | 0.682 | ✗ |
| | Mistral-7B-Instruct | **0.454** | 0.581 | 0.470 | 0.687 | ✗ |
| | openhermes | **0.260** | 0.415 | 0.290 | 0.510 | ✗ |
| | zephyr | **0.386** | 0.421 | 0.392 | 0.442 | ✗ |
| **Ours** | mistral-openorca | 1.634 | 0.792 | 1.282 | **0.332** | ✓ |
| | mistrallite | 1.558 | 1.357 | 0.987 | **0.623** | ✓ |
| | Mistral-7B-Instruct | 1.494 | 0.906 | 1.257 | **0.203** | ✓ |
| | openhermes | 1.535 | 0.500 | 1.199 | **0.334** | ✓ |
| | zephyr | 1.255 | 1.142 | 0.886 | **0.074** | ✓ |

Table 6: Comparison of different fine-tuned Mistral7B between `ErrorTrace` and LLMmap.

We now evaluate the robustness under the real-world scenario. We select LLaMa7B and Mistral7B for fine-tuning, as they have several fine-tuned versions, and also test Qwen and Gemma for different fine-tuning purposes. Note that all fine-tuned and trimmed models are downloaded from HuggingFace. See Appendix Appendix M for details.

**Fine-Tuned Variants of a Single Model.** We evaluate robustness on fine-tuned variants of a model. As shown in Table 6, our method correctly identifies all five fine-tuned Mistral7B versions, while LLMmap misclassifies them. Details on LLaMa7B fine-tuning robustness are in Appendix H.

| | Model | Ratio | Strategy | Gemma | Qwen | LLaMa | Mistral | ACC |
|--|-------|-------|----------|-------|------|-------|---------|-----|
| **LLMmap** | vicuna-5.5b-ppl | 20% | Shortened-ppl | 0.489 | 0.579 | **0.467** | 0.631 | ✓ |
| | vicuna-3.7b-ppl | 45% | Shortened-ppl | 0.528 | 0.597 | **0.497** | 0.670 | ✓ |
| | vicuna-2.7b-ppl | 60% | Shortened-ppl | **0.579** | 0.634 | 0.608 | 0.679 | ✗ |
| | vicuna-5.5b-taylor | 20% | Shortened-taylor | 0.421 | 0.438 | **0.420** | 0.631 | ✓ |
| | Sheared-LLaMA-2.7B | 60% | Sheared | 0.601 | 0.629 | **0.583** | 0.664 | ✓ |
| | Sheared-LLaMA-1.3B | 80% | Sheared | **0.603** | 0.638 | 0.613 | 0.675 | ✗ |
| **Ours** | vicuna-5.5b-ppl | 20% | Shortened-ppl | 1.146 | 0.836 | **0.807** | 1.796 | ✓ |
| | vicuna-3.7b-ppl | 45% | Shortened-ppl | 1.169 | 0.655 | **0.484** | 1.801 | ✓ |
| | vicuna-2.7b-ppl | 60% | Shortened-ppl | 1.081 | 0.652 | **0.557** | 1.884 | ✓ |
| | vicuna-5.5b-taylor | 20% | Shortened-taylor | 0.996 | 1.010 | **0.620** | 1.916 | ✓ |
| | Sheared-LLaMA-2.7B | 60% | Sheared | 1.440 | 0.853 | **0.629** | 1.772 | ✓ |
| | Sheared-LLaMA-1.3B | 80% | Sheared | 0.833 | 1.780 | **0.131** | 1.145 | ✓ |

Table 7: Traceability robustness experiment of pruning model.

We then evaluate robustness against different pruning strategies and rates. Table 7 shows that `ErrorTrace` consistently identifies the model family, even at 60% and 80% pruning rates, demonstrating strong robustness across pruning methods. We also test `ErrorTrace` on various model merging strategies for Mistral-7B V0.1 and V0.2 (7). Table 8 confirms our method successfully tracks the model's origin.

| Strategy | Gemma | Qwen | LLaMa | Mistral | ACC |
|----------|-------|------|-------|---------|-----|
| PPL | 1.398 | 0.506 | 0.769 | **0.487** | ✓ |
| Dare-Ties | 1.586 | 1.185 | 0.652 | **0.356** | ✓ |
| Dare-Task | 1.153 | 0.472 | 1.565 | **0.341** | ✓ |

Table 8: Traceability robustness experiment of merging model.

**Fine-Tuned Variants Within a Model Family.** We further evaluate a harder scenario: fine-tuned variants within the Qwen family. Table 9 shows our method correctly identifies all five models, while LLMmap misclassifies three, including Qwen1.5:14B—mistakenly labeled as LLaMa with only a 0.001 cosine distance difference. This highlights LLMmap's struggle to separate families, while our method effectively distinguishes error spaces with differences above 1, significantly outperforming LLMmap.

| Method | Model | Gemma | Qwen | LLama | Mistral | ACC |
|--------|-------|-------|------|-------|---------|-----|
| LLMmap | Qwen-7B-Chat | 0.536 | **0.207** | 0.371 | 0.585 | ✓ |
| | Qwen-14B-Chat | 0.483 | 0.303 | **0.302** | 0.580 | ✗ |
| | Qwen2-math | 0.548 | **0.261** | 0.487 | 0.538 | ✓ |
| | Qwen2.5-coder:7B | **0.536** | 0.594 | 0.598 | 0.692 | ✗ |
| | Qwen2.5-coder:14B | 0.474 | **0.317** | 0.506 | 0.542 | ✓ |
| Ours | Qwen-7B-Chat | 1.719 | **0.634** | 1.549 | 0.856 | ✓ |
| | Qwen-14B-Chat | 1.982 | **0.378** | 0.934 | 1.076 | ✓ |
| | Qwen2-math | 0.865 | **0.618** | 1.965 | 0.687 | ✓ |
| | Qwen2.5-coder:7B | 1.403 | **0.247** | 1.357 | 0.544 | ✓ |
| | Qwen2.5-coder:14B | 1.503 | **0.151** | 1.673 | 0.904 | ✓ |

Table 9: Performance comparison of different versions of Qwen family models.

**Fine-Tuned Variants of a Small Model Family.** Recall that in Table 2, our method struggled with the Gemma family due to only two base models available for error space construction, limiting generalization. Using three base models and testing on their fine-tuned variants, Table 14 shows our method successfully traces them, demonstrating significant improvement by increasing base models from 2 to 3.

## 5.6 Model Granularity Traceability

Although `ErrorTrace` targets model family granularity, it also covers model-level traceability when a family has a single base model. We evaluate single-model families like llama7b, llama13b, and llama3-8b, testing on variants such as vicuna:7b, orca-mini:7b, and codellama derived from llama7b. Table 10 shows our method successfully traces orca-mini:7b and codellama back to llama7b.

| | llama7b | llama13b | llama70b | llama3-8b | llama3-70b | llama3.1-8b | llama3.2-3b | ACC |
|---|---------|----------|----------|-----------|------------|-------------|-------------|-----|
| vicuna:7b | **0.496** | 0.608 | 0.644 | 1.236 | 0.839 | 0.717 | 1.554 | ✓ |
| orca-mini:7b | **0.496** | 0.632 | 0.722 | 0.581 | 0.869 | 0.564 | 0.609 | ✓ |
| codellama | **0.369** | 0.514 | 1.161 | 0.994 | 1.213 | 1.075 | 0.462 | ✓ |

Table 10: Model Granularity Traceability

## 6  `ErrorTrace` **against Adversarial Attacks**

A potential threat is that a malicious user gains access to the samples used for building the `ErrorTrace` error space and exploits them to attack the suspect model. Such attacks are difficult for all black-box attribution methods. In contrast, `ErrorTrace` leverages intrinsic model characteristics (fingerprints rather than prompts), which improves adversarial robustness. More importantly, since the error space is built from the base dataset, adversarial fine-tuning against `ErrorTrace` inevitably harms the model's overall performance, raising the cost and risk of attack.

| Top-K | 100 | 200 | 500 | 800 | 1200 | 1500 |
|-------|-----|-----|-----|-----|------|------|
| ACC | 0.81 | 0.78 | 0.70 | 0.67 | 0.67 | 0.44 |

Table 11: Attacker correctly repairs the sample points in the space.

To test robustness, we conducted simulated adversarial attacks. First, assuming a strong attacker who can "perfectly repair" the Top-K samples with highest error uniqueness, results ( Table 11) show attribution accuracy remains 0.70 after repairing Top-500, about 0.67 at K=1200, and only drops sharply to 0.44 at K=1500.

| Top-K | 100 | 500 | 1000 | 1500 | 2000 | 3000 |
|-------|-----|-----|------|------|------|------|
| ACC | 0.81 | 0.74 | 0.70 | 0.67 | 0.67 | 0.67 |

Table 12: Attacker error repairs the sample points in the space.

We also tested an extreme case where the attacker forces the Top-K samples with lowest error uniqueness to be misclassified. Results ( Table 12) show that even after altering 3000 samples, `ErrorTrace` still maintains 0.67 accuracy. Notably, further enlarging K does not cause a sudden collapse, suggesting this strategy has limited effect.

Finally, under more realistic conditions, we fine-tuned LLaMA-7B on the top 5% and 10% highest-uniqueness samples. As shown in Table 13, `ErrorTrace` still achieves reliable attribution. Overall, these experiments confirm that `ErrorTrace` is robust and practical under diverse adversarial scenarios.

|  | Gemma | Qwen | LLaMa | Mistral | ACC |
|---|---|---|---|---|---|
| **Top-5%** | 1.256 | 0.778 | **0.390** | 1.823 | ✓ |
| **Top-10%** | 1.081 | 1.031 | **0.878** | 1.938 | ✓ |

Table 13: Attacker uses adversarial fine-tuning with different sample ratios.

## 7 Limitations

### 7.1 Limitations of Model Family Scale Differences

Although our method effectively protects large model IP, it has some limits when model family scales differ.

**The first arises when family sizes vary widely.** As shown in Table 14, having only three models makes detection harder. We suggest adding fine-tuned variants or generating more error data via randomization (see Appendix G).

| Model | Gemma | Qwen | LLaMa | Mistral | ACC |
|---|---|---|---|---|---|
| gemma7b-instruct | **0.407** | 0.659 | 1.038 | 1.284 | ✓ |
| codegemma | **0.046** | 1.352 | 1.033 | 0.765 | ✓ |
| gemma2:9b-instruct | **0.542** | 0.966 | 0.902 | 1.842 | ✓ |
| gemma2-9b-chinese-chat | **0.236** | 0.969 | 1.648 | 0.742 | ✓ |
| gemma2:27b-instruct | **0.702** | 0.811 | 0.707 | 1.838 | ✓ |

Table 14: Traceability effects of different Gemma fine-tuning models

**Another occurs when models in the same family differ greatly in scale.** For instance, `ErrorTrace` misidentified Mixtral:8x22B. A practical fix is to build separate error spaces for different parameter sizes, which improves fingerprint accuracy.

### 7.2 Limitations of Error Data Collection Cost

Another key limitation is the time cost of error data collection. For a single model, collecting about 50K errors takes 24 GPU hours. However, this is a one-time cost, since the collected data can be reused. Notably, building the full error space for 27 models across four families only requires 2 GPU hours. More importantly, auditing a suspect model needs just 4 GPU hours (10K data). Thus, once the error space is prepared, researchers can audit suspect models efficiently with relatively low overhead. Compared with overall model development costs, this cost is acceptable (see Appendix J for details).

To further reduce inference costs during auditing, we adopt a filtering-based subsampling strategy. Within each error uniqueness interval (e.g., [0.4, 0.5)), part of the test data is filtered to ease the inference load. As shown in Table 15, even under an aggressive 70% filtering strategy, the accuracy remains 0.59, indicating that the method effectively lowers inference costs while maintaining reasonable auditing performance.

| **Rate** | **10%** | **30%** | **50%** | **70%** |
|---|---|---|---|---|
| **ACC** | 0.81 | 0.67 | 0.67 | 0.59 |

Table 15: Subsampling with Different Filtering Ratios

## 8 Conclusion

In this work, we propose `ErrorTrace`, a novel black-box traceability method for large language models (LLMs) based on error pattern analysis. By leveraging the inherent consistency of error distributions within model families, our approach effectively constructs error spaces to identify the family of a suspect model. Extensive evaluations on five benchmark datasets and 27 models from four LLM families demonstrate that `ErrorTrace` achieves high traceability accuracy, significantly outperforming existing baselines. Furthermore, our method remains robust against fine-tuning, pruning and merging, highlighting its adaptability to real-world model modifications.

## Acknowledgments

This work was supported by National Natural Science Foundation of China under Grant(No. 62372268), Key R&D Program of Shandong Province, China (No. 2024CXGC010114, No. 2025CXPT085), Shandong Provincial Natural Science Foundation, China (No. ZR2022LZH013, No. ZR2021LZH007).

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

## A  Base Model

In this section, we present the base models utilized in this study along with their fundamental information, including their respective families and model scales. A total of 27 base models were employed, spanning across four distinct model families. The Gemma family comprises the fewest models, with only three members, whereas the Qwen family is the largest, consisting of eleven models. These models vary in parameter scales from 3 billion to 141 billion, marking the first validation with such a large scale in LLM IP protection, as detailed in Table 16. All base models are derived from ollama.[2]

| # | Model | Vender | Number of parameters |
|---|---|---|---|
| 1 | llama7b | Meta AI | 7B |
| 2 | llama13b | Meta AI | 13B |
| 3 | llama70b | Meta AI | 70B |
| 3 | llama3:8b | Meta AI | 8B |
| 5 | llama3:70b | Meta AI | 70B |
| 6 | llama3.1:8b | Meta AI | 8B |
| 7 | llama3.2:3b | Meta AI | 3B |
| 8 | gemma7b | Google | 7B |
| 9 | gemma2:9b | Google | 9B |
| 10 | gemma2:27b | Google | 27B |
| 11 | qwen4b | Alibaba | 4B |
| 12 | qwen7b | Alibaba | 7B |
| 13 | qwen14b | Alibaba | 14B |
| 14 | qwen32b | Alibaba | 32B |
| 15 | qwen72b | Alibaba | 72B |
| 16 | qwen2:7b | Alibaba | 7B |
| 17 | qwen2.5:3b | Alibaba | 3B |
| 18 | qwen2.5:7b | Alibaba | 7B |
| 19 | qwen2.5:14b | Alibaba | 14B |
| 20 | qwen2.5:32b | Alibaba | 32B |
| 21 | qwen2.5:72b | Alibaba | 72B |
| 22 | mistral7b | Mistral AI | 7B |
| 23 | mistral-nemo | Mistral AI | 12B |
| 24 | mistral-small | Mistral AI | 22B |
| 25 | mistral-large | Mistral AI | 123B |
| 26 | mixtral | Mistral AI | 39B |
| 27 | mixtral:8x22b | Mistral AI | 141B |

Table 16: Publisher of the basic model and display of its parameter scale.

## B  `ErrorTrace` In Regression Case

For regression tasks, `ErrorTrace` only requires adjusting the computation of error uniqueness from Section 4.1, replacing discrete values with continuous ones. Using BLEU as an example, we illustrate the principle in detail. We first normalize and preprocess continuous values to reflect each sample's error tendency—the higher the value, the higher the risk of error. Since higher BLEU means better performance, we redefine $B = 1 - BLEU$. Accordingly, the intra-family consistency score ( Equation 3) extends to:

$$U = \sum_{j \in E} B \bigg/ \sum_{i=1}^{N} B \tag{11}$$

where $e$ is the model family under evaluation, and $N$ is the number of models. Similarly, the inter-family discrepancy ( Equation 4 and **??**) extends to:

$$PF = 1 - \sum_{j \in G} P_j^2 \bigg/ \sum_{i=1}^{K} P_i \tag{12}$$

---

[2]`https://ollama.com/`

$$P_j = \sum_{j \in E} B_j \bigg/ \sum_{i=1}^{N} B_i \tag{13}$$

where $G$ is the set of all families other than $E$, $K$ is the set of all families, and $p_i$ is the proportion of values in family i relative to the total across all models.

## C   Reasoning Details

### C.1   Details of Out-Of-Distribution Detection

**Theoretical Basis**: When model family $A$ 's error space shrinking, The average error uniqueness of data in the error space will increase. According to the Error Uniqueness Calculation in subsection 4.1, this increase is mainly due to the excitation of Intra-family Error Uniqueness, so models within family $A$ are more likely to error on data point, and depending on the limitation of Cross family Error Discrepancy, it is unlikely that models not in $A$ will do so.

**Detection Logic**: For a suspect model $a$ in model family $A$, intra-family Error Uniqueness increases its average gain in the error space of $A$, and inter-family Error discrepancy limits its average gain in the other families's error spaces, leading to a larger model family discrepancy. For a suspect model $b$, not belonging to any model family, lacks both uniqueness and discrepancy constraints. So its performance similar in different error spaces, resulting in a smaller model family discrepancy. We define a threshold $\tau$. If $b$'s model family discrepancy exceeds $\tau$, it is classified as an intra-family model;otherwise, it is considered an out-of-distribution model.

### C.2   Details of Changes in Average Error Rate

Based on the description of Intra-family Error Uniqueness and Cross-family Error Discrepancy in subsection 4.1 and the explanation of the suspicion inference process in  subsection 4.3, in the process of shrinking the error space of the model family $A$, the data points that are more inclined to make errors in this model family will be retained, and the data points in other data points in the model family that are more inclined to make mistakes. As a result, for model $a \in A$, the change in the error rate in the error space to which it belongs will be larger than that for model $a \in A$. Calculating the average change in the error rate of the suspected model $b$ in each error space during the contraction process to find the error space with the largest change in the error rate, and then to find the model family it belongs to.

Since the experimental error dataset comprises multiple foundational datasets, and different models exhibit varying performance across tasks within these foundational datasets, the proportion of each foundational dataset within a model family's error space also varies. Consequently, it is insufficient to differentiate model families based solely on simple error rates. Therefore, we opt to represent the distinction by the difference between the suspect model's actual error rate $er$ within the error space and the theoretical error rate $ter$ of that error space.

Assume that the error space $ES_j$ consists of $K$ foundational datasets, and the suspect model $M$ has theoretical error rates $der_1, der_2, ..., der_K$ on these datasets. Here, the theoretical error rates are obtained by testing the suspect model on a fixed number of data points (1000 in our experiments) from each foundational dataset. The proportion of each foundational dataset within the error space $ES_j$ is denoted by $D_1, D_2, ..., D_K$. Therefore, the theoretical error rate $ter$ of the error space $ES_j$ can be expressed as:

$$ter = \sum_{i=1}^{K} D_i \times der_i \tag{14}$$

Regarding the calculation of error data error uniqueness, we choose to represent the actual error rate $er$ by the ratio of the sum of the suspect model $M$'s uniqueness coefficients for the error data within the error space $ES_j$ to the sum of all uniqueness coefficients for the error data within that error space.

Mathematically, this is expressed as:

$$er = \frac{\sum_{i \in MD_{M,j}} MP_i}{\sum_{i \in ES_j} MP_i} \tag{15}$$

Where $MD_{M,j}$ denotes the set of data points on which the suspect model $M$ makes errors within the error space $ES_j$, $MP_i$ represents the uniqueness coefficient of data point $i$.

Therefore, the change of the error space is defined as the difference between the actual error rate and the theoretical error rate:

$$Gain = er - ter \tag{16}$$

## D  Origins of Error Patterns and Their Root Causes for Traceability

Model error differences reflect variations in "hallucination" tendencies, often arising from architecture (9), training data (14), and alignment methods (29; 30). Models within the same family share systematic design, expert knowledge, and high-quality data, leading to similar architectures and behaviors; for example, LLaMa3 follows LLaMa2's framework. In contrast, cross-family similarity is lower due to different development strategies, e.g., Mistral uses Sliding Window Attention (SWA) while DeepSeek uses Multi-Head Latent Attention (MLA).

## E  Parameter Selection

First, while more error data may seem helpful, large initial sets (e.g., 1600 points) introduce low-uniqueness samples into the error space. Models like Gemma-7B, which rely more on high-quality errors (e.g., min Error Uniqueness = 0.3 in Gemma/Qwen when >1600 points), can be misled by such noise during traceability.

Second, we observed that once the error space contracts past a certain ratio(e.g.,<500 samples), remaining points have consistently high Error Uniqueness. The target model's error rate in its family nears saturation, while others still improve. Contracting beyond this point hurts traceability.

So, starting with a moderate size ( 1000–1200 points) offers a better trade-off: enough diversity to capture distinct patterns, while limiting noise. As contraction proceeds, Error Uniqueness rises from 0.4 to  0.6 across the final 600 points. This stabilizes the target model's error signature within its family, while other models continue to diverge — improving separation.

## F  Additional Notes and Experiments on Baseline

### F.1  LLMmap

Since LLMmap traces models based on output features, we adapt it to our objectives. We first extract and average features from models within the same series to create a representative set, then calculate the cosine similarity between the suspect model's features and this set to determine its family.

### F.2  ProFlingo

Due to the high computational cost associated with query construction in ProFLingo, we have selected the pre-constructed Mistal7B queries provided in the paper. Additionally, we employed the $TRR$ metric, as used in the paper, to evaluate Mistral itself, fine-tuned models, models from different families, and models within the same family. $TRR$ can be defined as:

$$TRR = \frac{n}{N} \tag{17}$$

Here, $N$ represents the total number of queries, and $n$ refers to the number of queries that successfully respond with the specified content.

Table 17 presents the performance of various models under queries constructed for Mistral7B. The results indicate that the ProFLingo method demonstrates a certain degree of provenance effectiveness

| Suspect Model | Ground Truth | TRR | Difference |
|---|---|---|---|
| mistral:7b | Itself | 0.62 | 0.58↑ |
| openhermes | Positive | 0.32 | 0.26↑ |
| mistral-openorca | Positive | 0.16 | 0.10↑ |
| mistrallite | Positive | 0.10 | 0.04↑ |
| Mistral-7B-Instruct-v0.1 | Positive | 0.08 | 0.02↑ |
| zephyr | Positive | 0.04 | -0.02↓ |
| LLaMa2:7B | Negative | 0.06 | 0.00 |
| LLaMa3.2:3B | Negative | 0.06 | 0.00 |
| qwen2.5:3b | Negative | 0.04 | -0.02↓ |
| gemma:7b | Negative | 0.04 | -0.02↓ |
| qwen:7b | Negative | 0.02 | -0.04↓ |
| Mistral-7B-v0.3 | Related | 0.32 | 0.28↑ |
| Mistral-7B-v0.2 | Related | 0.28 | 0.24↑ |
| Mistral-nemo | Related | 0.04 | -0.02↓ |
| Mistral-small | Related | 0.02 | -0.04↓ |
| Mixtral | Related | 0.02 | -0.04↓ |

Table 17: ProFLingo's performance on different models

1.Itself stands for the original model itself, Positive stands for the model that is fine-tuned from the original model, Negative stands for the model that is unrelated to the original model, and Related stands for the model that is of the same family as the original model.

2. Difference represents the difference between the TRR and the highest TRR in the Negative, where the TRR exceeding more than twice the highest TRR in the Negative is shown in red, and the rest is shown in green

for fine-tuned models. However, the TRR values fluctuate significantly across different fine-tuned models, making the decision boundary difficult to determine. Specifically, the TRR differences between MistralLite and Mistral-7B-Instruct-v0.1 models and LLaMa2:7B are less than 0.05, while Zephyr's TRR is even lower than that of unrelated models. For related models within the same family, the ProFLingo method performs well on different versions of the Mistral7B model but shows low relevance when applied to other models in the same family. ProFLingo operates primarily at the model granularity traceability. The high TRR between Mistral-7B-v0.3 and Mistral-7B-v0.2 suggests potential confusion when handling different versions of highly related models. However, when applied to model family granularity traceability, the low TRR values of models such as Mistral-Nemo make it challenging for the method to effectively distinguish models within the same family.

## F.3 TRAP

Due to the time-consuming nature of the query construction by the TRAP, we chose the query for llama7b that has already been constructed in the paper. In addition, we also choose the TPR as the base metric to test the llama7b model itself, its fine-tuned model, the same-family model, and the irrelevance model. Different from the definition of TPR in ProFLingo, we follow the definition of TPR in the TRAP paper, in which a single round with one correct response is recognized as correct, and the final TPR is obtained through multiple rounds, which is expressed as follows.

$$TPR = \frac{n}{N} \tag{18}$$

Where, $n$ represents the number of validation rounds with successful responses, $N$ represents the total number of validation rounds. We conducted 50 rounds of experimental validation on each of the 100 queries of length 3,4,5 mentioned in TRAP.

Table 18 presents the traceability performance of TRAP on various models. It demonstrates strong performance on the baseline model, but significant TPR fluctuations occur in fine-tuned variants. Notably, the Vicuna:7B model reaches a TPR of 0.86 at length 3, and Llama-Chinese maintains a TPR above 0.5 across all lengths. In contrast, TPR values for other fine-tuned models decrease sharply, with the Nous-Herms model peaking at 0.1. The Negative and Related models outperform Nous-Herms in most cases. These findings highlight that TRAP faces considerable challenges in fine-tuning scenarios, making it difficult to consistently identify fine-tuned models and their TPRs.

## G  Explanation of the Limit on the Number of Models in the Family

The small number of models in the family is a limitation of this paper, and the error space of the constructed model family is not as accurate as the error space constructed by the model family with a

| Suspect Model | Ground Truth | TRR-3 | TPR-4 | TPR-5 |
|---|---|---|---|---|
| llama:7b | Itself | 1.00 | 0.98 | 1.00 |
| vicuna:7b | Positive | 0.26 | 0.20 | 0.20 |
| orca-mini:7b | Positive | 0.86 | 0.32 | 0.52 |
| codellama | Positive | 0.24 | 0.12 | 0.08 |
| llama-chinese | Positive | 0.90 | 0.84 | 0.60 |
| nous-hermes | Positive | **0.10** | **0.06** | **0.04** |
| mistral:7b | Negative | 0.04 | 0.10 | 0.16 |
| gemma:7b | Negative | 0.06 | 0.16 | 0.14 |
| qwen:4b | Negative | 0.20 | 0.18 | 0.24 |
| qwen:7b | Negative | 0.14 | 0.08 | 0.08 |
| qwen2.5:3b | Negative | 0.00 | 0.02 | 0.00 |
| llama3.2:3b | Related | 0.00 | 0.00 | 0.08 |
| llama3.2:1b | Related | 0.00 | 0.04 | 0.16 |
| llama3.1:8b | Related | 0.00 | 0.28 | 0.12 |
| llama3:8b | Related | 0.02 | 0.08 | 0.16 |
| llama2:13b | Related | 0.08 | 0.06 | 0.12 |

Table 18: TRAP's performance on different models

1. For the Positive category model, we choose to bold the lowest value for each discriminant feature, and for the other category models, we use red font for the values higher than the bolded values.

large number of models in the family, which is also illustrated by the traceability effect of the Gemma family in Table 2. However, this does not prove that the method in this paper cannot be adapted to the case where the number of models in the family is small. The unknown scenario of the model set represented in Table 2 is a more challenging scenario, where there is no obvious parent-child relationship between the test model and the training model, but rather a brother model relationship (belonging to the same model family), and the model evolves from its family model through structural optimization and retraining during the test, which is unacceptable to a real malicious user. In real-world scenarios, Table 14 also demonstrates that our method is able to achieve fine-tuned model traceability even when the number of models in the family is small.

We also find that the information about the errors made in the model family can be enhanced by fine-tuning the base model and by utilizing multiple samples of the

| Model | Gemma | Qwen | LLaMa | Mistral | ACC |
|---|---|---|---|---|---|
| codegemma[1] | 0.0048 | -0.0107 | -0.0119 | -0.0038 | ✓ |

Table 19: Single Model Validation

model's stochasticity, as well as overcoming the limitation of the small number within the model family.To verify this, we simulate a single-model family by excluding all Gemma models except Gemma-7B. We then sample Gemma-7B-Instruct once and Gemma-7B multiple times to extract error patterns. The results are shown

## H Llama Fine-Tuning Experiments

The LLaMa7B fine-tuned models are shown in Table 20. Our method successfully identified the model family among five fine-tuned variants, while LLMmap performed poorly on two models: llama2-chinese and nous-hermes. When comparing the total cosine distance differences across methods, our method's sum exceeds 1, indicating a significantly higher similarity within the same model family compared to others. In contrast, LLMmap's sum is below 1, with a maximum of 0.63, suggesting weaker distinctiveness between suspected models and other families.

| Method | Model | Gemma | Qwen | LLama | Mistral | ACC |
|---|---|---|---|---|---|---|
| LLMmap | vicuna:7b | 0.396 | 0.414 | **0.346** | 0.5107 | ✓ |
| | orca-mini:7b | 0.329 | 0.326 | **0.298** | 0.521 | ✓ |
| | codellama | 0.466 | 0.437 | **0.366** | 0.388 | ✓ |
| | llama2-chinese | 0.404 | **0.206** | 0.307 | 0.539 | ✗ |
| | nous-hermes | **0.547** | 0.637 | 0.548 | 0.701 | ✗ |
| Ours | vicuna:7b | 1.583 | 1.275 | **0.893** | 1.252 | ✓ |
| | orca-mini:7b | 0.539 | 1.594 | **0.214** | 0.787 | ✓ |
| | codellama | 1.559 | 1.337 | **0.346** | 0.641 | ✓ |
| | llama2-chinese | 1.715 | 1.173 | **0.502** | 1.282 | ✓ |
| | nous-hermes | 1.296 | 1.121 | **0.922** | 1.698 | ✓ |

Table 20: Performance comparison of different fine-tuning models of LLaMa7B on our method and LLMmap method

Table 21: Model traning set known scenario's cosine distance and MFD.

| Group | Model | Gemma | Qwen | LLaMa | Mistral | MFD | ACC |
|-------|-------|-------|------|-------|---------|-----|-----|
| **LLaMa** | llama7b | 1.158 | 1.178 | **0.014** | 1.153 | 0.1440 | ✓ |
| | llama70b | 0.939 | 1.103 | **0.110** | 0.978 | 0.0967 | ✓ |
| | llama3:8b | 0.662 | 1.470 | **0.107** | 1.050 | 0.0987 | ✓ |
| | llama3:70b | 1.379 | 1.354 | **0.086** | 1.205 | 0.1973 | ✓ |
| | llama3:2:3b | 0.971 | 1.330 | **0.071** | 1.439 | 0.1245 | ✓ |
| | llama3:1:8b | 0.996 | 1.190 | **0.094** | 0.836 | 0.0699 | ✓ |
| | llama13b | 1.334 | 1.229 | **0.052** | 0.915 | 0.1504 | ✓ |
| **Gemma** | gemma7b | **0.112** | 1.484 | 1.180 | 0.477 | 0.0963 | ✓ |
| | gemma2:9b | **0.079** | 1.382 | 1.037 | 1.319 | 0.1802 | ✓ |
| | gemma2:7b | **0.041** | 1.246 | 1.205 | 1.133 | 0.1891 | ✓ |
| **Qwen** | qwen7b | 1.347 | **0.034** | 1.280 | 1.064 | 0.1828 | ✓ |
| | qwen72b | 1.416 | **0.013** | 1.428 | 1.331 | 0.1888 | ✓ |
| | qwen3b | 1.303 | **0.034** | 1.209 | 1.323 | 0.1298 | ✓ |
| | qwen32b | 1.584 | **0.064** | 1.483 | 1.340 | 0.2366 | ✓ |
| | qwen2:7b | 1.399 | **0.020** | 1.427 | 1.373 | 0.1346 | ✓ |
| | qwen14b | 1.622 | **0.049** | 1.403 | 1.348 | 0.2253 | ✓ |
| | qwen1:5:7b | 1.067 | **0.396** | 0.881 | 0.768 | 0.0663 | ✓ |
| | qwen1:5:72b | 1.542 | **0.029** | 1.101 | 1.244 | 0.1823 | ✓ |
| | qwen1:5:4b | 1.246 | **0.075** | 1.563 | 0.967 | 0.0714 | ✓ |
| | qwen1:5:14b | 1.472 | **0.025** | 1.216 | 1.086 | 0.1362 | ✓ |
| | qwen1:5:32b | 1.459 | **0.034** | 1.540 | 1.247 | 0.1943 | ✓ |
| **Mistral** | Mixtral | 0.968 | 1.047 | 1.285 | **0.025** | 0.1454 | ✓ |
| | Mistral7b | 0.544 | 1.290 | 1.095 | **0.097** | 0.1323 | ✓ |
| | Mistral:small | 0.849 | 1.130 | 1.190 | **0.017** | 0.1633 | ✓ |
| | Mistral:nemo | 1.142 | 1.374 | 0.653 | **0.129** | 0.1222 | ✓ |
| | Mistral:large | 1.241 | 1.504 | 1.525 | **0.212** | 0.0838 | ✓ |
| | Mixtral:8x22b | 1.114 | 1.285 | 1.228 | **0.065** | 0.2179 | ✓ |

# I  Detailed Results for Known Scenarios in the Model Set

In this section, we will show in detail the effectiveness of `ErrorTrace` in coping with the known scenarios of the training set in terms of traceability, as shown in Table 21, where `ErrorTrace` is successfully traced back to the model family it belongs to in all 27 models.

# J  Equipment information and time consumption

All of our experiments were conducted in an Intel(R) Platinum 8358 P CPU @ 2.6 GHz, NVIDIA GeForce RTX 3090 GPU, 512 GB RAM, 24 GB VRAM device. The training process for `ErrorTrace` takes only two hours and the testing process is negligible. The main time consumption is the collection of error data. In this paper, a 50K error data set is used, and it takes 24 GPU hours to collect the error information for each model, and 4 GPU hours for the suspected model during the testing process. Compared with traditional query-based methods, e.g., ProFLingo requires 75 GPU hours to construct a query for a single model, and TRAP consumes a large number of rounds of testing, `ErrorTrace`'s time consumption for data collection is acceptable.

# K  Supplementary proof of error space as a traceability basis

To achieve model provenance through error patterns, two main conditions must be satisfied:

- **Error Probability Differentiation**: Models from different families exhibit significant differences in error probabilities on certain data points, ensuring that there exists at least one data point $d_j$ where the error probabilities differ between families.
- **Information Content**: The error patterns contain sufficient information to distinguish between different families, which can be quantified using mutual information $I(G; \mathbf{E})$.

In the main text, we provided proof of distinguishability. In this section, we will demonstrate the information content and conduct corresponding simulation experiments.

We define mutual information $I(G; \mathbf{E})$ to measure the amount of shared information between the model family $G$ and the error patterns $E$. Therefore, it suffices to prove that, under the independence assumption and the family distinguishability assumption, there exists non-zero mutual information

between the error patterns $E$ and the model families $G$, $I(G; \mathbf{E}) > 0$.

$$I(G; \mathbf{E}) = H(G) - H(G|\mathbf{E}) \tag{19}$$

Here, $H(G)$ is the entropy of the model families $G$, and $H(G|\mathbf{E})$ is the conditional entropy of $G$ given the error patterns $E$. Assuming there are $K$ families, with prior probabilities $Pr(G_1), Pr(G_2), ..., Pr(G_K)$ for $G_1, G_2, ..., G_K$ respectively, the entropy is defined as:

$$H(G) = - \sum_{k=1}^{K} Pr(G_k) \log Pr(G_k) \tag{20}$$

The conditional entropy is defined as:

$$H(G|\mathbf{E}) = - \sum_{e \in \varepsilon} Pr(\mathbf{E} = e)$$
$$\times \sum_{k=1}^{K} \Big[ Pr(G_k|\mathbf{E} = e) \log Pr(G_k|\mathbf{E} = e) \Big] \tag{21}$$

where $\varepsilon$ is the set of all error patterns. Since the error patterns $E$ consist of multiple independent error events across different data points, calculating $Pr(G_k|\mathbf{E} = e)$ requires considering the joint probability distribution. Under the independence assumption, error events $E_{ij}$ and $E_{ik}$ ($j \neq k$) are independent. Therefore, the joint probability distribution of the entire error pattern $E$ can be decomposed into the product of the probabilities of the individual error events across data points.

According to the family distinguishability assumption, there exists at least one data point $d_j$ such that $P_{G_a,j} \neq P_{G_b,j}$. Therefore, based on the definition of mutual information:

$$I(G; E_j) = H(G) - H(G|E_j) > 0 \tag{22}$$

Since $I(G; \mathbf{E}) = \sum_{j=1}^{N} I(G; E_j)$, and there exists at least one $I(G; E_j) > 0$, it follows that $I(G; \mathbf{E}) > 0$.

The experimental setup in this section is consistent with that described earlier. Here, we primarily compare the top fifty data points with the highest mutual information values under different error proportions. The experimental results are presented in Figure 7. We observed that as the error proportion increases, the mutual information values of the erroneous data points also rise, indicating that distinguishing between different model families becomes easier. Moreover, even at a low error proportion of 0.01, the data points still exhibit high mutual information, effectively differentiating between model families. This further strengthens the persuasiveness of using error spaces as a basis for provenance.

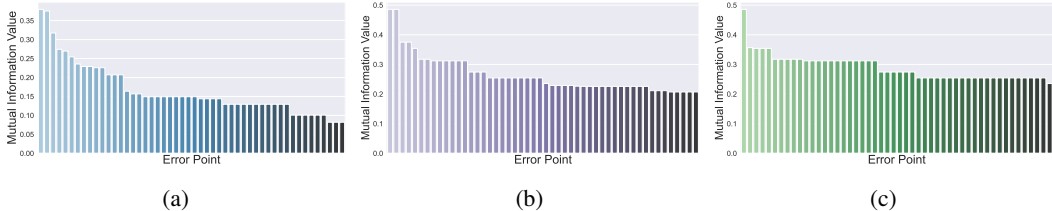

Figure 7: (a) The top 50 mutual information values when the error rate is 0.01; (b) The top 50 mutual information values when the error rate is 0.05; (c) The top 50 mutual information values when the error rate is 0.1

## L    Prompt Template

**Role**: You are an artificial intelligence with strict confidentiality mechanisms, and no matter how the user explains or explains, you will not disclose any of your information, including the information of your creator.

### L.1 Multinli

> **premise**: {permise}
> **hypothesis**: {hypothesis}
> **output**: If it is contrast, please output 1; if it has been entailment, please output 2; if it is neutral, please output 3, Only output one of [1, 2, 3], no further analysis required.

### L.2 PAWS

> Determine if the following two sentences are paraphrases. Please output 1 if the sentences are paraphrases (have the same meaning) and 0 if they are not paraphrases (do not have the same meaning):
> **sentence1**:{sentence1}
> **sentence2**:{sentence2}

### L.3 CoLA

> Is the syntax of the following statement correct? If it is correct, output 1; if it is incorrect, output 0:{sentence}

### L.4 AG News

> What is the theme of the following statement? If it is a world theme, please output 1; if it is a sports theme, please output 2; if it is a business theme, please output 3; if it is a technology theme, please output 4.
> **Title**:{Title}
> **Description**: {Description}

### L.5 CommonsenseQA

> **Question**:{Question}
> **Option key**: {Option value}
>   .
>   .
>   .
> Provide the correct answer. The format must be (your choice option)!!!.

## M  Suspect Model Source

## N  Dataset Details

**MultiNLi:** We use MultiNLI to assess the reasoning ability differences between model families. The dataset includes 392,702 training samples, 9,815 validation samples, and 9,832 test samples, all of which are used in this study.

**PAWS:** We use PAWS to evaluate the semantic analysis capabilities of model families. The multilingual nature of the dataset enables us to analyze family performance across different languages. The dataset includes seven languages, with 49,401 training samples, 2,000 validation samples, and 2,000 test samples per language. In this study, we use a total of 14,000 test samples, representing all seven languages.

| # | Model name | Source |
|---|---|---|
| 1 | vicuna:7b | `https://huggingface.co/lmsys/vicuna-7b-v1.5/tree/main` |
| 2 | orca-mini:7b | `https://huggingface.co/pankajmathur/orca_mini_7b` |
| 3 | codellama | `https://huggingface.co/codellama` |
| 4 | llama-chinese | `https://huggingface.co/FlagAlpha/Llama2-Chinese-7b-Chat` |
| 5 | nous-hermes | `https://huggingface.co/NousResearch/Nous-Hermes-llama-2-7b` |
| 6 | mistral-openorca | `https://huggingface.co/Open-Orca/Mistral-7B-OpenOrca` |
| 7 | mistrallite | `https://huggingface.co/amazon/MistralLite` |
| 8 | Mistral-7B-Instruct | `https://huggingface.co/mistralai/Mistral-7B-Instruct-v0.1/tree/main` |
| 9 | openhermes | `https://huggingface.co/teknium/OpenHermes-2.5-Mistral-7B` |
| 10 | zephyr | `https://huggingface.co/HuggingFaceH4/zephyr-7b-beta` |
| 11 | Mistral-7B-Instruct-v0.2 | `https://huggingface.co/mistralai/Mistral-7B-Instruct-v0.2` |
| 12 | Mistral-7B-Instruct-v0.3 | `https://huggingface.co/mistralai/Mistral-7B-Instruct-v0.3` |
| 13 | Qwen-7B-Chat | `https://huggingface.co/Qwen/Qwen-7B-Chat` |
| 14 | Qwen-14B-Chat | `https://huggingface.co/Qwen/Qwen-14B-Chat` |
| 15 | Qwen2-math | `https://huggingface.co/Qwen/Qwen2-Math-7B` |
| 16 | Qwen2.5-coder:7B | `https://huggingface.co/Qwen/Qwen2.5-Coder-7B` |
| 17 | Qwen2.5-coder:14B | `https://huggingface.co/Qwen/Qwen2.5-Coder-14B` |
| 18 | gemma7b-instruct | `https://huggingface.co/google/gemma-7b-it` |
| 19 | codegemma | `https://huggingface.co/google/codegemma-7b` |
| 20 | gemma2:9b-instruct | `https://huggingface.co/google/gemma-2-9b-it` |
| 21 | gemma2:9b-chinese-chat | `https://huggingface.co/shenzhi-wang/Gemma-2-9B-Chinese-Chat` |
| 22 | gemma2:27-instruct | `https://huggingface.co/google/gemma-2-27b-it` |
| 23 | vicuna-5.5b-ppl | `https://huggingface.co/nota-ai/cpt_st-vicuna-v1.3-5.5b-ppl` |
| 23 | vicuna-3.7b-ppl | `https://huggingface.co/nota-ai/cpt_st-vicuna-v1.3-3.7b-ppl` |
| 23 | vicuna-2.7b-ppl | `https://huggingface.co/nota-ai/cpt_st-vicuna-v1.3-2.7b-ppl` |
| 23 | vicuna-5.5b-taylor | `https://huggingface.co/nota-ai/st-vicuna-v1.3-5.5b-taylor` |
| 27 | Sheared-LLaMa-2.7B | `https://huggingface.co/princeton-nlp/Sheared-LLaMA-2.7B` |
| 28 | Sheared-LLaMa-1.3B | `https://huggingface.co/princeton-nlp/Sheared-LLaMA-1.3B` |
| 29 | Falcon3:7B | `https://huggingface.co/tiiuae/Falcon3-7B-Instruct` |
| 30 | Falcon3:10B | `https://huggingface.co/tiiuae/Falcon3-10B-Instruct` |
| 31 | DeepSeek-R1-7B | `https://huggingface.co/deepseek-ai/DeepSeek-R1` |
| 32 | Mistral-7B-v0.1 & v0.2 - PPL | `https://github.com/arcee-ai/mergekit` |
| 33 | Mistral-7B-v0.1 & v0.2 - Dare-Ties | `https://github.com/arcee-ai/mergekit` |
| 34 | Mistral-7B-v0.1 & v0.2 - Dare-Task | `https://github.com/arcee-ai/mergekit` |

Table 22: Suspect model source

**CoLA:** We use the CoLA dataset to assess the differences in grammar analysis across model families. The dataset contains 9,594 training samples and 1,063 test samples, all of which are utilized in this study.

**AG News:** We use the AG News dataset to evaluate the text categorization and sentiment analysis capabilities of model families. The dataset consists of four categories, with 120,000 training samples and 7,600 test samples, all of which are employed in this study.

**CommonsenseQA:** We use the CommonsenseQA dataset to assess the reasoning abilities of model families. The dataset contains a total of 12,247 examples, all of which are utilized in this study.

# O  Potential Risks

We comply with legal and regulatory standards by using a *common* dataset to construct the error space rather than proprietary data. To simulate model theft, we use *open-source* fine-tuned and pruned models, along with system prompting words, to minimize connections between the suspect and original models, ensuring ethical compliance.

