# OpenReview forum: "ErrorTrace: A Black-Box Traceability Mechanism Based on Model Family Error Space"
_NeurIPS.cc/2025/Conference — NeurIPS 2025 spotlight_

### Official Review · Reviewer_ykmK · 2025-06-30

**Clarity:** 4
**Significance:** 3
**Originality:** 3
**Rating:** 4
**Confidence:** 3

**Summary:**

This paper presents a novel black-box traceability framework called ErrorTrace for identifying the source model family of a suspect large language model (LLM), with the goal of protecting intellectual property (IP). The key idea is to leverage the error uniqueness patterns of LLM families—specific prompts where only certain model families make high-confidence but incorrect predictions.

**Questions:**

Please answer the questions in the Weaknesses.

**Ethical Concerns:**

["NO or VERY MINOR ethics concerns only"]

**Final Justification:**

Rating score (4) is maintained, with clarity score increased to 4.

**Limitations:**

yes

**Quality:**

3

**Strengths And Weaknesses:**

Strengths:
1. The proposed method is interest, leveraging the error distribution patterns of large language models to identify the source model family of a suspect model under black-box conditions.
2. The empirical results are strong and convincing, and the authors provide code for reproducibility, which facilitates adoption in the LLM security and IP protection community.

Weaknesses / Questions:
1. The approach assumes that the model’s error space exhibits a structurally stable and distinguishable pattern. However, in scenarios where the suspect model undergoes aggressive fine-tuning (e.g., on data distributions highly similar to those used for error reference computation), it is unclear whether such structure persists. How robust is ErrorTrace under these conditions?
2. The method seems more focused on family-level attribution. Can it differentiate between models within the same family, or protect the IP of individual models?
3. As model capacity increases and training data scales up, the error diversity across families might diminish. Has the authors considered whether the discriminative power of error spaces vanishes under this condition?
4. The core assumption behind this method is that the error space is family-specific. What are the underlying causes of such error divergence? Are they primarily due to architectural biases, training data differences, optimization dynamics, or other factors?
5. The method relies on predefined thresholds (e.g., for cosine similarity) to attribute model IP. However, the generalization and stability of these thresholds are not well justified. For instance, Table 3 shows considerable overlap in cosine similarities between matched and unmatched model families.

---

> ### Author Rebuttal · Authors · 2025-07-31
>
> We are very grateful for your efforts on our work. Many of the comments you raised are important, and we will address these issues one by one:
>
> > 1.Concerns About Adversarial Attacks
>
> Any black-box traceability method faces challenges if attackers know the detector’s query data—often impossible to fully prevent. Still, we have studied this and provided explanations.
>
> Moreover, ErrorTrace’s strength is in using a model’s inherent traits, like identifying a person by their fingerprint, not a spoken phrase. Adversarial fine-tuning to fool ErrorTrace harms the model’s overall performance.
>
> To handle black-box API costs and rate limits, we use filtering-based subsampling (see table below), within each Error Uniqueness range (e.g., [0.4, 0.5)), some data is filtered during testing, cutting inference overhead. Even with aggressive filtering (70%), accuracy remains reach 0.59.
>
> | Top-K | 100  | 200  | 500  |
> | :---- | :--- | :--- | :--- |
> | ACC   | 81%  | 78%  | 70%  |
>
> We also test a strong adversarial case where the attacker forced incorrect predictions on the Top-K data points with the lowest error uniqueness. Even after modifying the Top-1000 points, ErrorTrace still achieved 0.7 attribution accuracy (see table below).
>
> | Top-K | 100  | 500  | 1000 |
> | :---- | :--- | :--- | :--- |
> | ACC   | 81%  | 74%  | 70%  |
>
> Additionally, under a real attacker scenario, we fine-tuned LLaMA-7B with the top 5% and 10% highest Error Uniqueness data. ErrorTrace still successfully performed traceability (see table below).
>
> |         | Gemma | Qwen  | LLaMa | Mistral |
> | :------ | :---- | :---- | :---- | :------ |
> | Top-5%  | 1.256 | 0.778 | 0.390 | 1.823   |
> | Top-10% | 1.081 | 1.031 | 0.878 | 1.938   |
>
> > 2.Concerns About Model Identification
>
> We ran model-level tests in Table 10 (Section 5.6), but the presentation may have caused confusion. In Section 5.6, we built error spaces for seven LLaMA models and tested three fine-tuned LLaMA-7B variants. All fine-tuned models were correctly traced back to their originals. For these same-company models, ErrorTrace reliably linked fine-tuned models to their sources.
>
> > 3.Concerns About Model Scale
>
> Your question is highly relevant. We have considered this situation and believe that the discriminative ability of the error space is unlikely to diminish as model capacity increases. The reason of this will be discussed in Response 4.
>
> Furthermore, we argue that the poor traceability performance of large-scale models is mainly due to the imbalance in model size distribution rather than the loss of discriminative error signals. To address this, ErrorTrace (Equations (3)–(5) in Section 4.1) adjusts the weights of erroneous data from large models to preserve fingerprint reliability. A practical fix is building separate error spaces when parameter gaps exist, improving fingerprint accuracy.
>
> > 4.Similarity of error patterns
>
> This question is both interesting and important. As discussed in Section 1, we attribute the family-specific characteristics of error spaces to differences in model architecture, training data, and optimization strategies, which collectively shape the inductive biases and failure modes of each model family.
>
> - Model architectures: The model architecture defines the computational graph and information flow, which results in distinct processing strengths and systematic weaknesses. Different families often adopt distinct design choices. For instance, Mistral uses Sliding Window Attention (SWA) [1], while DeepSeek employs Multi-head Latent Attention (MLA) [2].
>
>
> - Training data: Differences in data management, filtering, and sampling can lead to distinct failure modes, even when overall corpus quality is similar. For proprietary model datasets, such differences in error patterns may be even more significant.
>
>
> - Optimization dynamics: Variations in training schedules, learning rate warmups, and regularization introduce distinct inductive biases, which manifest as variations in error distributions.
>
>
> Furthermore, our empirical results in Sections 3 and 5 show that traceability based on model family error patterns performs well in both simulated and real environments, supporting this explanation.
>
> We will elaborate on these reasons in detail in the final version.
>
> > 5.Threshold definition
>
> We did not apply any threshold to the cosine distance. Instead, we used an MFD-based threshold to determine whether a suspicious model is OOD. This threshold was derived from both theoretical and empirical analyses.
>
> - Cosine distance:  ErrorTrace identifies the suspected model’s family based on the minimum cosine distance. As shown in Table 3, this “minimum value” strategy consistently works, in all 23 successful cases, the model with the lowest cosine distance always belongs to the correct family.
>
>
> - MFD threshold for OOD detection:  As described in Appendix B and Section 5.4, we select the optimal threshold (0.025) by designating the LLaMA family as OOD and the other families (Gemma, Qwen, Mistral) as in-distribution, then analyzing the distribution of model family differences. In practice, all 58 in-distribution models tested (including perturbed models such as fine-tuned or pruned versions) have MFD values above 0.025 (minimum = 0.0314), while all 3 OOD models (Table 5) have values below 0.025 (maximum = 0.0167). These results confirm both the generalization and stability of the chosen threshold.
>
> [1] Mistral 7B
>
> [2] DeepSeek-V2 Technical Report

---

> > ### Comment · Area_Chair_bdP4 · 2025-08-06
> > **Discussion required.**
> >
> > Dear Reviewer ykmK,
> >
> > Please read carefully through the authors' responses and check if they address all your concerns.
> >
> > With kind regards,
> >
> > Your AC

---

> > ### Comment · Reviewer_ykmK · 2025-08-09
> > **Response to Authors**
> >
> > Thank you for the elaborated response. The rebuttal has addressed some of my concerns, especially about the discriminative ability related to model scale. In terms of the risk of adversarial attacking and aggressive fine tuning, theoretical discussion is not thoroughly enough and experiments (i.e., tuned on attacking purpose) can be included. But overall, this is still an interesting paper with meaning findings. Rating score (4) is maintained, with clarity score increased to 4.

---

> > > ### Author Response · Authors · 2025-08-09
> > > **Acknowledgment of Reviewer’s Feedback**
> > >
> > > We sincerely thank the reviewer for the thorough and constructive feedback. We are encouraged by the recognition of our findings and the increase in the clarity score. We acknowledge the valuable suggestions on strengthening our discussion of adversarial attacks and aggressive fine-tuning, and we plan to address these points by adding more in-depth theoretical analysis and targeted experiments in our future revision.

---

### Official Review · Reviewer_tvyD · 2025-07-02

**Clarity:** 2
**Significance:** 3
**Originality:** 3
**Rating:** 3
**Confidence:** 3

**Summary:**

This paper introduces ErrorTrace, a novel black-box traceability mechanism for Large Language Models (LLMs) that addresses the issue of unauthorized model derivatives. Unlike existing methods, ErrorTrace analyzes unique error patterns within model families by mapping their distinct error spaces. This approach allows for robust intellectual property (IP) protection without needing access to internal model parameters or specific query responses. ErrorTrace demonstrates high accuracy in identifying the origins of suspect models, even when fine-tuned, pruned, or merged, outperforming baselines and showcasing broad applicability

**Questions:**

1. Explain why the baseline methods perform poorly in this paper's setting.
2. Discuss the effect of the number of samples.

**Ethical Concerns:**

["NO or VERY MINOR ethics concerns only"]

**Final Justification:**

After reading the rebuttal, I still believe the computational overhead introduced by this paper is a significant drawback compared to existing methods. Therefore, I have decided to maintain my score.

**Limitations:**

Although the authors acknowledge the additional overhead from dataset collection and claim this cost is "manageable" (about 24 GPU hours per model), I still consider this a significant drawback compared to existing methods like LLMmap.

**Quality:**

2

**Strengths And Weaknesses:**

Strengths:

1. The idea of using the failure behaviors of the LLMs as model fingerprints is novel.
2. This paper studies black-box LLM fingerprinting, which is practical in real world.
3. The experiments in this paper are comprehensive.

Weaknesses:

1. The performance of the baseline (e.g., LLMmap) is much lower in this paper (about 60% accuracy) compared to what was reported in its original paper (about 80%). I understand this might be due to different experimental setups, but it may be better for the authors to explain why LLMmap performs so poorly under their specific settings.
2. In this paper's experiments, the proposed methods need 50K data points, which represents a very high cost.
3. I also wonder if the accuracy improvement comes from using significantly more data than the baseline. The paper's method uses 50K data points, while LLMmap only needs 8 queries. Would LLMmap's performance improve if it were also allowed to use more data? On the other hand, does the proposed method still work well when only using 100 or fewer samples?
4. The paper's writing could be improved. Specifically, Section 4.3 omits some important details, which may reduce the paper's readability and make it hard to understand the method completely.

---

> ### Author Rebuttal · Authors · 2025-07-31
>
> We sincerely appreciate your recognition of the novelty, practicality, and effectiveness of our work. We will address each of your concerns point by point:
>
> > 1.Explanation of the LLMmap effect
>
> We first emphasize that we have peformed LLMmap under its original setting and achieved a success rate of 92.59% (indeed higher than 85.18% presented in paper). We fully acknowledge LLMmap’s effectiveness in its original scenario.
>
> However, the key reason for the performance gap lies in the difference between our threat model and that of LLMmap.  Our method, ErrorTrace, is designed for IP protection where attackers actively attempt to conceal model provenance (see Section 2.1).  To simulate such adversarial concealment, we apply a system prompt instructing suspect models to hide identifying information, representing a realistic and challenging theft scenario.
>
> By contrast, LLMmap targets identification of unknown models without such concealment, often relying on manually crafted queries that directly request model-specific information (e.g., manufacturer names).  While effective in its setting, these queries are easy to defend against when the attacker intends to obscure their model identity.
>
> Our experimental results validate that this concealment strategy significantly reduces LLMmap’s accuracy under our scenario, highlighting the robustness and practical relevance of our approach for IP protection.
>
> We will add a detailed discussion in Section 5.1 about the differing experimental setups and threat models between our work and LLMmap.  Furthermore, Section 5.2 will include analyses of LLMmap’s performance under various configurations to clarify its strengths and limitations.  We believe these additions will greatly enhance the rigor and clarity of the comparison.
>
> > 2.Cost concerns
>
> Thank you for highlighting this important limitation, which we address in Section 6 (Limitations). We acknowledge that ErrorTrace requires a relatively large number of queries and labeled data during the error space construction phase. However, this is a one-time offline cost: the 50K samples used to build the error space enable efficient detection across many models. After data screening in Section 4.2, the suspect model detection phase uses only ~10K compressed samples, requiring just 4 GPU hours—significantly less than the resources typically needed for model training.
>
> In contrast to adversarial query methods such as ProFLingo and Trap, which demand hours of GPU time per crafted query and multiple queries per model, our approach is considerably more practical. As shown in Table 4 (Section 5.3), even reducing the construction dataset to 20K samples still achieves 70% accuracy, providing a cost-performance trade-off.
>
> Additionally, as noted in our response to Response 3, we introduce a filtering-based subsampling strategy during inference that reduces reasoning overhead with minimal impact on accuracy.
>
> We will clarify these points and include a discussion on cost trade-offs and practical solutions in the final version.
>
> > 3.Concerns about the fairness of LLMmap
>
> As noted in Response 1, LLMmap depends on eight expert-crafted queries, which most users are not equipped to create. The original paper also shows that performance does not improve beyond eight queries. In contrast, ErrorTrace automates this process by selecting high-quality queries from standard benchmark datasets, shifting the effort from manual design to automated computation.
>
> To validate this, we increased the number of queries in LLMmap and ran experiments (see table below). Results showed that as more **ordinary queries** were added, LLMmap’s performance declined—highlighting its reliance on expert-designed high-quality queries.
>
> |      | 8    | 20   | 50   | 75   | 100  |
> | ---- | ---- | ---- | ---- | ---- | ---- |
> | ACC  | 59%  | 63%  | 59%  | 59%  | 56%  |
>
> Furthermore, we also evaluated how data volume affects ErrorTrace (see table below). During inference on 10K samples, we filtered each unique error interval (e.g., [0.4, 0.5]) to reduce sample size. Results show that ErrorTrace maintains stable performance as the filtering ratio increases—achieving 59% accuracy even with 70% of the data removed.
>
> | Rate | 10%  | 30%  | 50%  | 70%  |
> | ---- | ---- | ---- | ---- | ---- |
> | ACC  | 0.81 | 0.67 | 0.67 | 0.59 |
>
> > 4.Writing problems
>
> We sincerely thank you for raising this point. Due to space constraints, some details were omitted from the main text. However, Appendix B provides further explanation, including the theoretical basis and detection logic for OOD detection, as well as a deeper analysis of error space contraction. We will include a full description in the final version.
>
> > 5.The explanation of LLMmap
>
> We have provided a detailed discussion in our Response 1.
>
> > 6.The impact of query quantity
>
> We provided a detailed explanation in Response 3.
>
> > 7.Limitations
>
> As noted in Response 2, we would like to further clarify the similarities and differences between our method and LLMmap. Both rely on high-quality queries, but LLMmap depends on expert-crafted prompts, limiting its generalizability. In contrast, ErrorTrace automatically identifies high-quality queries from large datasets—effectively shifting the reliance from human expertise to machine computation.

---

### Official Review · Reviewer_dntv · 2025-07-21

**Clarity:** 2
**Significance:** 2
**Originality:** 2
**Rating:** 2
**Confidence:** 3

**Summary:**

The paper introduces ErrorTrace, a black box method for tracing IP of LLMs. The proposed method follows three steps: computing error scores, constructing an error space using graph-based techniques and inferring the source model based on the error rates. The method is evaluated on a synthetic datasets and compare it to the logistic regression and random forest.  While this paper addresses an important problem, the core contribution are heuristics and system oriented. The method lacks theoretical depth, relies on strong and untested assumptions, and offers little in the way of novel algorithm insights. Its empirical evaluation is limited to synthetic settings, leaving open questions about its robustness and applicability in real-world scenarios. Overall, the work feels more a prototype than a principled solution.

**Questions:**

1. How robust is the method to realistic model variation? All evaluations assume fairly large and distinguishable differences in error rates (≥0.2). What happens if two models only differ slightly or only in out-of-distribution regions?

2. Why should error patterns be consistent within model families? Without formal or empirical justification, the assumption that error vectors form distinguishable manifolds seems fragile—especially under fine-tuning.

3. How scalable is the method to modern LLMs with >100B parameters? The current method assumes full access to error vectors on curated datasets. It's unclear whether the method scales to foundation models like LLaMA or GPT in realistic deployment settings.

4. Is there a lower bound on traceability? Under what conditions (e.g., number of data points, degree of error divergence) is model attribution statistically guaranteed?

5. Is there any interpretability or causal attribution? The method identifies families based on aggregate errors but offers no insight into why a model belongs to a particular family. Could it misattribute due to spurious correlations?

**Ethical Concerns:**

["NO or VERY MINOR ethics concerns only"]

**Final Justification:**

My concerns on theory still remain, so I will keep the score.

**Limitations:**

Another important limitation that isn’t addressed is the vulnerability of the method to adaptive attacks. Since ErrorTrace relies on identifying distinct error patterns, it’s not hard to imagine an adversary fine-tuning a model in a way that deliberately smooths or scrambles its error distribution to evade detection. For example, a malicious actor could inject noise, overfit on a set of known test prompts, or train against a proxy ErrorTrace-like detector. Without any analysis or discussion of how the method holds up under these kinds of threat-aware manipulations, it’s hard to assess its reliability in adversarial settings. A method meant for model fingerprinting or IP protection should at least consider the possibility of being targeted itself. Including even a basic discussion or preliminary experiment on this front would make the claims much more convincing.

**Paper Formatting Concerns:**

I do not have any concerns on the paper formatting.

**Quality:**

1

**Strengths And Weaknesses:**

Strengths

1. Black-box setting relevance: The goal of tracing derivative LLMs without internal access is practically important, especially given open-weight model proliferation.
2. Systematic pipeline: The proposed method is well-structured and comprises clearly defined phases—error uniqueness calculation, error space construction, and model inference.
3. Empirical validation: Some synthetic experiments are performed to support the feasibility of using error patterns for model identification, with visual analysis (e.g., error trajectory plots) to aid interpretability.

Weaknesses

1. Shallow Theoretical Insight: The probabilistic framework is overly simplistic, relying on strong assumptions like complete observability of binary error vectors and independence of errors across data points. The “family distinguishability” assumption is merely stated without analysis of when or why it holds, nor how it scales with model capacity or distributional shift. No formal characterization of identifiability, information bounds, or generalization is given.

2. Lack of Novel Learning or Algorithmic Techniques: All core components—error rate computation, graph-based neighborhood construction, uniqueness scoring—use simple heuristics or weighted frequency counts. The graph construction (Section 4.2) uses BFS and CDF-weighted distances, but does not introduce any principled method grounded in information theory, statistical learning, or robust optimization.

3. Overreliance on Synthetic Experiments: The experiments focus entirely on synthetic toy datasets with hand-designed error distributions, which may not reflect the complexity of real LLM behavior. No real-world benchmark or practical LLM traceability evaluation is presented.

4. Missing Discussion of Attack Resilience: The method may be trivially circumvented by adversaries adding noise or modifying their error profiles (e.g., via targeted fine-tuning or prompt filtering), but this is not discussed or tested. There is no robustness analysis against domain shift, few-shot adaptation, or pruning-induced error alignment.

---

> ### Author Rebuttal · Authors · 2025-07-31
>
> We appreciate your recognition of our work's practical value. We address your questions below, one by one:
>
> > 1.Theoretical Explanation
>
> We agree that stronger theoretical grounding would improve generality and interpretability, practical effectiveness is paramount in model IP protection.  However, most black-box methods lack formal theory and rely on empirical validation. Our key contribution lies in a novel traceability perspective, supported by experiments—a strength noted by Reviewers 3iae, tvyD, and ykmK. We acknowledge our earlier explanation may have been unclear and now provide a clearer theoretical overview.
>
> First, our assumptions are not overly strong but are based on observed patterns and empirical results.
>
> - Regarding the independence assumption, model predictions are independent when context is reset before each generation—an operational setup feasible in real detection.
>
> - Regarding the model family distinguishability assumption, it is grounded in the intuition that models from the same family share similar errors due to common architecture, training data, and optimization. Figure 1(a) supports this, showing higher error overlap within families. While we lack a formal proof across all capacities and shifts, experiments on 61 real LLMs in Section 5 show this assumption holds well in practice.
>
> - Regarding the completeness assumption, as long as LLM responses can be collected and correctness judged, error events can be recorded.
>
>
> Second, regarding model scale and families with different size distributions, theoretical analysis is limited due to challenges in comparing activations across scales. However, we empirically study scale effects on family separation(Table 3).
>
> Third, regarding identifiability and generalization, we study how model count, error distribution, data size, and key error proportion affect family attribution. Results are shown in Table 1 and Figure 2. Furthermore, appendix H provides an information-theoretic perspective on how error proportions influence distinguishable information.
>
>
> > 2.Innovation of the Algorithm
>
> Our main contribution is a novel traceability perspective and framework that contrasts with adversarial query-based methods. Built on fundamental principles, our design simplifies understanding and application, boosting practicality. Experiments show this straightforward approach achieves strong results, with accuracy up to 0.85(Table 3).
>
> Specifically, our approach leverages statistical principles to ensure distinguishability between model families—unlike brute-force query methods. Section 4.1 defines intra-family consistency incentives and inter-family distinguishability penalties to identify data that separates model families. To account for large-scale models, we incorporate model performance into both calculations.
>
>
> > 3.Concerns over Synthetic Experiments
>
> We appreciate the reviewer’s concern about reliance on synthetic experiments. To clarify, our experiments mainly use **real, general-purpose benchmark datasets** and analyze error behaviors of **authentic LLMs (Section 5)**.
>
> Furthermore, the “synthetic” aspect refers only to an exploratory simulation of error distributions to test traceability feasibility. However, the main results come from error spaces built directly from **real model outputs, without handcrafted or artificial error injection**.
>
> > 4.Concerns about countering attacks
>
> We have presented pruning experiments in Section 5.5. Next, we will discuss targeted fine-tuning.
>
> In fact, it is indeed an inherent challenge for any black-box traceability method if an attacker gains knowledge of the query data used by the detector. To validate ErrorTrace under adversarial conditions, we run a test assuming a strong attacker—one who perfectly fixes all Top-K data points with the highest error uniqueness in the target space. Even after altering the Top-500 data points, the attribution accuracy stays as high as 0.7(see table below).
>
> | Top-K | 100  | 200  | 500  |
> | ----- | ---- | ---- | ---- |
> | ACC   | 81%  | 78%  | 70%  |
>
> We also test a strong adversarial case where the attacker forced incorrect predictions on the Top-K data points with the lowest error uniqueness. Even after modifying the Top-1000 points, ErrorTrace still achieved 0.7 attribution accuracy (see table below).
>
> | Top-K | 100  | 500  | 1000 |
> | ----- | ---- | ---- | ---- |
> | ACC   | 81%  | 74%  | 70%  |
>
> Additionally, under a real attacker scenario, we fine-tuned LLaMA-7B with the top 5% and 10% highest Error Uniqueness data. ErrorTrace still successfully performed traceability (see table below).
>
> |         | Gemma | Qwen  | LLaMa | Mistral |
> | ------- | ----- | ----- | ----- | ------- |
> | Top-5%  | 1.256 | 0.778 | 0.390 | 1.823   |
> | Top-10% | 1.081 | 1.031 | 0.878 | 1.938   |
>
> > 5.Robustness of real models
>
> First, the assumption of an error rate difference ≥0.2 applies only to **exploratory experiments(Section 3)**. **Formal experiments (Section 5) use real data (e.g., MultiNLI) and real models (e.g., LLaMa-7B)**, with measured error rates that vary from 0 to 1 across model families.
>
> Second, **In Section 5.5**, we thoroughly test robustness under subtle differences, **including fine-tuning variants and models from advanced pruning and merging**. Results show that despite small error rate variations, traceability performance remains stable (see Table 7).
>
> > 6.Similarity of error patterns
>
> Regarding your concern about the similarity of error patterns within model families, we have already addressed this in detail in Question 1 under "Model Family Distinguishability."
>
> Furthermore,**as for fine-tuning, our experimental results demonstrate that fine-tuning does not significantly affect the traceability results(Section 5.5).**
>
> > 7.Concerns about model size
>
> In our main experiments, we conducted validation on **real models such as LLaMa and successfully achieved traceability even under scenarios involving model fine-tuning, pruning, and merging**. We did not assume access to the dataset's error information; instead,**all error information was automatically selected by our method in real benchmark**.
>
> Furthermore, we experimented with **models exceeding 100B parameters, such as Mixtral:8x22B (141B)，Mistral-large(123B)**. Although traceability was not successful in this case, we believe this is reasonable given the complexity of the unknown model scenarios and the significant scale differences within the Mistral family. Nevertheless, we successfully achieved precise traceability on the Qwen:72B model.
>
> > 8.Lower limit problem
>
> In Table 1 (Section 3), we have demonstrated this point through simulations: with 5,000 data points and an error divergence of 0.05, a random forest detector achieves 80% accuracy. **However, such controlled conditions are rare in real-world settings**.
>
> In practice, we cannot directly control error differences of real models—especially large black-box LLMs, whose internal logic is inaccessible. But **if a model family exhibits sufficiently unique errors on even a few key data points, traceability is statistically feasible**. Our ability to trace all modified models in the robustness experiments on fine-tuning, pruning, and merging (Section 5.5) further supports this point.
>
> Furthermore, regarding data volume, Table 4 shows that with just 20K data points, our method achieves around 70% accuracy—11 points above the baseline.
>
> > 9.Interpretability
>
> Thank you for raising important questions about interpretability and causal attribution.
>
> However, given the black-box setting and large intra-family differences (e.g., layers, neurons), fine-grained interpretability at the neuron or layer level remains challenging. Still, the traceability accuracy of up to 0.85 on real LLMs (Section 5.2) demonstrates the feasibility of our method.
>
> We acknowledge that false correlations may cause misattribution. To reduce this risk, our approach considers intra-family similarities and inter-family differences(Section 4.1), and uses multiple diverse datasets to minimize such effects(Section 5.1).
>
> > 10.Limitations
>
> We provided the answer in Question 2.

---

> > ### Comment · Area_Chair_bdP4 · 2025-08-06
> > **Discussion required.**
> >
> > Dear Reviewer dntv,
> >
> > Please read carefully through the authors' responses and check if they address all your concerns.
> >
> > With kind regards,
> >
> > Your AC

---

> > ### Comment · Reviewer_dntv · 2025-08-08
> > **Post-Rebuttal Response**
> >
> > The rebuttal fails to address the most critical weaknesses of the work:
> >
> > 1. **Lack of Theoretical Grounding**
> >    The authors’ position that “most black-box methods lack formal theory” is not a justification for also omitting it. This reasoning is self-defeating—high-impact research should raise the standard, not defer to the lowest common denominator. The method hinges on strong assumptions (independence, family distinguishability, completeness) yet still offers no formal analysis of identifiability, information bounds, or conditions for failure. Without such grounding, the claimed generality and robustness remain speculative.
> >
> > 2. **Limited Novelty of Algorithmic Contributions**
> >    The core components—error computation, graph-based distances, frequency weighting—are standard heuristics with no principled advancement in statistical learning or optimization. The “traceability perspective” alone does not constitute a substantive algorithmic contribution at the level expected for publication.
> >
> > 3. **Synthetic vs. Real-World Evaluation Gap**
> >    Despite additional results, the evaluation remains largely in controlled or curated settings. The real-model experiments still sidestep key challenges such as long-term model evolution, diverse adversarial strategies, and realistic noise conditions. As a result, the method’s robustness and utility in practical LLM traceability remain unproven.

---

> > > ### Author Response · Authors · 2025-08-08
> > > **An explanation for rebuttal's concerns**
> > >
> > > Thank you for your thoughtful comments. To address your concerns and clarify our methods and results, we provide the following explanations to improve understanding.
> > >
> > > > 1. Lack of Theoretical Grounding
> > >
> > > We fully agree—strong research should aim to push boundaries, not settle for minimal standards. Our approach was not a concession to the "lowest bar," but a response to **the real limitations of current interpretability tools in black-box settings**. **Without access to model internals**—like parameters or intermediate representations—standard interpretability methods are simply not feasible.
> > >
> > > Additionally, our focus is at the model family level. **Given that models in the same family often vary widely in size and architecture, existing interpretability methods struggle to support fair, consistent comparisons.** Therefore, we chose to analyze external behavioral patterns, which we believe offers a practical and scalable way forward for model provenance research.
> > >
> > > In Section 5, we evaluated our method on **27 real-world foundation LLMs across four model families, achieving a model family identification accuracy of 0.85**—demonstrating strong performance on original models. We further **tested 31 real modified LLMs (via fine-tuning, pruning, and merging), all of which were accurately traced back to their model families.** This confirms the robustness and generalizability of our approach in complex, real-world settings.
> > >
> > > > 2. The method is simple.
> > >
> > > We truly appreciate your insightful comments. We would like to emphasize that **our main contribution is a novel, effective framework for model traceability**, clearly distinct from existing provenance methods that rely heavily on adversarial queries. **Unlike those methods, which can be sensitive to query design and limited in scope, our approach uses the overall distribution of model behavior, providing a more robust and practical solution.**
> > >
> > > As you noted, our components are heuristic and basic. **Yet, their effective integration within our framework enabled us to achieve 85% traceability accuracy.** This result clearly shows the strength and practical value of our novel framework, highlighting its potential in complex provenance tasks.
> > >
> > > We greatly appreciate your valuable suggestions. With further optimization of the components, we expect improved traceability based on our proposed framework. **We will prioritize this direction in future work, continuously refining and enhancing our method.**
> > >
> > > > 3. Synthetic experiment
> > >
> > > We apologize for any confusion or concerns caused. **All LLMs in Section 5 were sourced from real-world settings, using only publicly available, authentic benchmark data. The experiments were neither controlled nor artificial but reflect real-world complexity and diversity.**
> > >
> > > In fact, long-term model evolution is hard to replicate in labs. Without continuous user optimization data for commercial LLMs, full reproduction isn’t feasible. However, in Section 5.5, we showed **ErrorTrace is robust against real-world modifications like fine-tuning, pruning, and merging, maintaining stable performance.**
> > >
> > > Regarding diverse adversarial strategies and real-world noise, we addressed this in our reply to comment (4). **Experiments confirm ErrorTrace is resilient to adversarial attacks, ensuring reliable model provenance.**

---

### Official Review · Reviewer_3iae · 2025-07-22

**Clarity:** 2
**Significance:** 4
**Originality:** 4
**Rating:** 5
**Confidence:** 4

**Summary:**

The paper proposes ErrorTrace, a black box lineage attribution scheme for LLMs. For each model family, the authors collect a large, labelled prompt set and record every reference model’s error on each prompt. They weight prompts by “error uniqueness”, where errors that are frequent within a family and are rare outside are assigned higher weights, and use a graph based pruning/BFS procedure to isolate a compact error space that serves as the family’s fingerprint. Given a suspect model, they measure how its weighted error rate drops as that space is progressively contracted; the family whose space shows the sharpest, out of distribution threshold exceeding drop is declared the origin. Experiments on 27 open source models across four families show 85% attribution accuracy in a leave one out setting, beating the strongest cosine similarity baseline (LLMmap, 59%). The method remains effective on fine tuned, pruned, and merged variants, and flags unseen families as OOD.

**Questions:**

**1. Cosine variance (Table 3)**: Within family cosine similarities can range up to three orders of magnitude. Could the authors clarify the mains sources of that spread? How does this square with the claim that family members share consistent error patterns?

**2. Contraction scaling trend (Figure 6b)**: Accuracy deteriorates when the start set is very large (≥ 1,600 nodes) and the final set is the minimum (500 nodes). Intuitively, retaining more nodes before contraction should only add information. Could the authors please provide quantitative guidance: what start/end sizes yield the best trade off, and why?

**3. Patch attack robustness**: Consider an adversary who (i) fine tunes on the top k high uniqueness prompts to eliminate those errors, or (ii) deliberately introduces random errors on k low uniqueness prompts to muddy the drop curve. As k increases, at what point does ErrorTrace fail, meaning it either mis-identifies the family or labels the model out of distribution under the current threshold τ? A plot of attribution accuracy versus k for both attack types would quantify the cost of evasion and help substantiate the method’s security claims.

**4. Minor Presentation Fixes**: In Figures 1a and 5a, it would be helpful to explicitly state in the captions what the numbers in the heat-maps represent. Additionally, there is a spelling error of “families” on line 154.

**Ethical Concerns:**

["NO or VERY MINOR ethics concerns only"]

**Final Justification:**

Thea authors' rebuttal(s) have addressed most my concerns, although a few still remain. Still, I believe the ideas introduced in the proposed work are solid enough to recommend this paper for acceptance, pending the inclusion of the experiments discussed in the rebuttal.

My remaining concern is with regard to this experiment: "Additionally, we extended Experiment 1 by increasing K (see table below). Accuracy stayed high at 67% for K = 1200, but dropped sharply to 44% at K = 1500." I don't think I have enough context to gauge whether this degree of adversarial robustness is sufficient in practice. To assuage concerns like these, I believe the authors would benefit from a more comprehensive examination of the robustness of their approach, considering red-teaming attacks beyond the ones discussed in this rebuttal. As an aside, I appreciate the authors' transparency in reporting the performance after red-teaming.

Because of the practicality of the proposed method and because of the authors' engagement during the rebuttal period, I will advocate for this paper and increase my score to 5.

**Limitations:**

Yes.

**Paper Formatting Concerns:**

Included at the end of the Questions Section.

**Quality:**

3

**Strengths And Weaknesses:**

Strengths:

- **Novel Perspective**: That models from the same families share the same peculiar mistakes is intuitive and original. Moreover, the proposed fingerprinting-scheme appears on its surface to be more practical and scalable than prior methods that require careful and extensive model modification.

- **Convincing Evaluation**: The authors evaluate their approach on 27 models over four model families, and include robustness checks on modified models via pruning, fine-tuning, and merging. The proposed method demonstrates significant empirical gain over prior work with generally limited failure modes.

- **Clarity**: The paper is clearly organized, with a motivating introduction, a well-defined threat model and method formulation section, and an accessible experimental section. The scope and the contributions are clear.

Weaknesses:

- **Data/Query Efficiency and Practicality**: A notable concern is the large number of queries and labeled data required to construct and use the error space. The approach relies on having a sizeable dataset of examples (the authors used 5 benchmark datasets, totaling tens of thousands of points) to probe each model and identify its error pattern. In fact, the experiments show that performance improves steadily as more data is used: ErrorTrace at 20k samples was ~70% accurate, rising to ~85% by 50k samples. Using 50k queries per suspect model (and similarly querying all reference models) is a substantial overhead. The authors note that collecting 50k data points for one model family costs about 24 GPU hours per model for inference, which is non-trivial. If the suspect model is only available via a pay-per-query API, this approach could be costly or even infeasible. While large organizations might afford this for important IP cases, it does limit practicality. The method’s efficiency could be an issue when scaling to many families or frequently testing new suspects. There is also a subtle point that the data used for errors needs to have ground-truth labels, which might not exist for every domain. In summary, the approach trades off fewer assumptions on the model for a heavy query requirement. This weakness doesn’t negate the approach’s viability, but it does raise questions about how it can be deployed operationally (especially against models that may rate-limit or charge for API calls).

- **Adversarial Evasion Considerations**: While ErrorTrace is more robust than prior methods to naive fine-tuning or pruning, the paper does not deeply explore adaptive adversarial scenarios. An attacker aware of this fingerprinting technique might intentionally try to mask or alter the error pattern of a stolen model. For instance, they could fine-tune the model on the very data points that are known “unique errors” for the source family, thereby “patching” those errors. Or they might inject noise into outputs for certain queries to confuse the trace. The current evaluation did not simulate an attacker who specifically aims to evade ErrorTrace (beyond standard fine-tuning for some task). It’s unclear how much leeway an adversary has to distort the error signature without significantly hurting model quality. This is a potential weakness because IP violators in practice might try to avoid detection. ErrorTrace’s reliance on inherent error patterns is clever, but if those patterns can be partially sanitized or hidden via additional training, the traceability could be undermined. The robustness to intentional fingerprint removal is not demonstrated, leaving a gap in the security analysis of the method.

- **Family vs. Model Identification**: By design, ErrorTrace focuses on identifying the model family, not the exact model. This is understandable (family-level is easier and arguably sufficient for many IP cases), but it is still a limitation in granularity. In scenarios where multiple organizations use the same base family (e.g., many clones of LLaMA exist), knowing a suspect “is a LLaMA-family model” might not conclusively prove it came from *your* instance of LLaMA. The paper does mention that if a family only contains one model, then effectively it performs model-level tracing, and they show examples of detecting specific fine-tunes (like Orca or CodeLlama) back to a single base model. However, for multi-model families, ErrorTrace doesn’t distinguish which family member was the source. This might be considered a weakness depending on the enforcement scenario. For example, if two companies train models on the same code base, more evidence would be needed to pin down theft. The paper could be stronger if it at least discussed this nuance or proposed how one might extend ErrorTrace to distinguish siblings in the same family (perhaps by incorporating some model-specific error features in addition to family-wide ones).

- **Behavior on Very Large or Adept Models**: In experiments, ErrorTrace sometimes failed to identify the largest models (e.g. “Mistral:large” 13B and a merged 22B model) as belonging to their family. The authors hypothesize that a dramatically larger model (or one formed by merging multiple models) might have a distinct error profile that diverges even from its smaller family members. This points to a general weakness: the assumption that a family’s errors are consistent may break down when models differ substantially in scale, training data, or architecture. In real-world terms, if an attacker took an open 7B model and scaled it up or merged it into a much larger model, would ErrorTrace still trace it to the original family? The paper’s results suggest this is challenging. The method might benefit from a way to adjust for model competency differences (since a very large and/or advanced model will make fewer errors overall, potentially making it harder to get a meaningful “error intersection” signature). At present, this remains an open problem for tracing state-of-the-art models if they are significantly improved over the references.

- **Extending to continuous tasks**: The entire pipeline hinges on a binary error indicator. In multiple choice or classification settings this is natural, but many high value LLM deployments involve continuous or graded outputs (e.g., proof generation, code synthesis, free form QA). How would ErrorTrace define “incorrect” and, by extension, error uniqueness in such domains? A proof checker’s pass/fail verdict could act as a binary flag, yet other tasks produce scores (BLEU, ROUGE, pass@k) rather than hard correctness. Without a clear extension path, ErrorTrace’s applicability seems limited to tasks with clear labels.

---

> ### Author Rebuttal · Authors · 2025-07-31
>
> We sincerely thank you for acknowledging the novelty, practicality, and effectiveness of our work. We deeply appreciate your time, effort, and thoughtful suggestions. Below, we respond to each of your concerns in detail.
>
> > 1.Concerns About Query Costs
>
> We thank the reviewer for highlighting this key issue. While ErrorTrace needs many queries and labeled data during error space construction, this is a one-time cost: the 50K samples are used only offline, and the error space can then detect many models efficiently. After Section 4.2, data screening during error space construction, suspect model detection uses only ~10K compressed samples(only 4GPU hours), which is a minimal cost compared to the millions of samples and extensive GPU hours typically required for model training.
>
> Furthermore, experiments in Section 5.2 have shown ErrorTrace built from open-source data reach 0.85 accuracy in unseen cases, showing minimal need for labeled data. When annotations are scarce, a small manual labeling effort plus semi-supervised expansion suffices.
>
> To handle black-box API costs and rate limits, we use filtering-based subsampling (see table below), within each Error Uniqueness range (e.g., [0.4, 0.5)), some data is filtered during testing, cutting inference overhead. Even with aggressive filtering (70%), accuracy remains reach 0.59.
>
> | Rate | 10%  | 30%  | 50%  | 70%  |
> | ---- | ---- | ---- | ---- | ---- |
> | ACC  | 0.81 | 0.67 | 0.67 | 0.59 |
>
> We will add a discussion on query costs in the final version, explaining their cause and possible solutions.
>
> > 2.Concerns About Adversarial Attacks
>
> Any black-box traceability method faces challenges if attackers know the detector’s query data—often impossible to fully prevent. Still, we have studied this and provided explanations.
>
> Moreover, ErrorTrace’s strength is in using a model’s inherent traits, like identifying a person by their fingerprint, not a spoken phrase. Adversarial fine-tuning to fool ErrorTrace harms the model’s overall performance.
>
> To validate ErrorTrace under adversarial conditions, we run a test assuming a strong attacker—one who perfectly fixes all Top-K data points with the highest error uniqueness in the target space. Even after altering the Top-500 data points, the attribution accuracy stays as high as 0.7(see table below).
>
> | Top-K | 100  | 200  | 500  |
> | ----- | ---- | ---- | ---- |
> | ACC   | 81%  | 78%  | 70%  |
>
> We also test a strong adversarial case where the attacker forced incorrect predictions on the Top-K data points with the lowest error uniqueness. Even after modifying the Top-1000 points, ErrorTrace still achieved 0.7 attribution accuracy (see table below).
>
> | Top-K | 100  | 500  | 1000 |
> | ----- | ---- | ---- | ---- |
> | ACC   | 81%  | 74%  | 70%  |
>
> Additionally, under a real attacker scenario, we fine-tuned LLaMA-7B with the top 5% and 10% highest Error Uniqueness data. ErrorTrace still successfully performed traceability (see table below).
>
> |         | Gemma | Qwen  | LLaMa     | Mistral |
> | ------- | ----- | ----- | --------- | ------- |
> | Top-5%  | 1.256 | 0.778 | **0.390** | 1.823   |
> | Top-10% | 1.081 | 1.031 | **0.878** | 1.938   |
>
> We will discuss this potential threat in detail in the final version, based on these findings.
>
> > 3.Concerns About Model Identification
>
> We ran model-level tests in Table 10 (Section 5.6), but the presentation may have caused confusion. In Section 5.6, we built error spaces for seven LLaMA models and tested three fine-tuned LLaMA-7B variants. All fine-tuned models were correctly traced back to their originals. For these same-company models, ErrorTrace reliably linked fine-tuned models to their sources, matching your scenario.
>
> > 4.Concerns About Model Scale
>
> Large-scale models remain a key challenge in this field. ProFLingo [1] was tested only on 7B models, and white-box Huref [2] up to 30B. In contrast, ErrorTrace success traces models up to 72B, a major advance.
>
> Building on this, as shown in Equations (3)–(5) (Section 4.1), model performance is explicitly incorporated into fingerprinting.
>
> Furthermore, among the households in Mistral, the scale distribution is uneven. Although large models have higher weights, the errors of many smaller models dominate. A practical fix is building separate error spaces when parameter gaps exist, improving fingerprint accuracy.
>
> We will detail this in the Mistral-large failure case (Section 5.2) in the final version.
>
> > 5.Extending to continuous tasks
>
> The error space focuses on analyzing model performance across tasks, not just classification. Thus, task type does not affect ErrorTrace.
>
> To extend to other tasks, the discrete {0,1} in Section 4.1 (Eq. 3–5) can be replaced with continuous values by adjusting the computation, without altering the core logic.
>
> We will elaborate on the method in Section 4 of the final version based on this response.
>
> > 6.Cosine variance (Table 3)
>
> The large differences in Table 3 arise from two main factors: ErrorTrace’s use of model performance and high error pattern similarity within families.
>
> First, as detailed in Section 4.1 and our answer to Question 4, lower-performance models less represent their family’s errors. So, Error Uniqueness downweights them, making their removal have little impact.
>
> Second, a stronger effect comes from error correlation. Like removing one of two correlated features in classic ML, dropping a model whose errors are highly correlated with others has minimal impact on the error space. A simple example is LLaMA-70B shares 0.64 error overlap with other LLaMa models—above the family average 0.6 in Figure 1(a)—while LLaMA3-70B has only 0.58.
>
> > 7.Contraction scaling trend (Figure 6b)
>
> We faced similar confusion in experiments. Analysis of the contraction process revealed insights omitted from the paper due to space.
>
> First, while more error data may seem helpful, large initial sets (e.g., 1600 points) introduce low-uniqueness samples into the error space. Models like Gemma-7B, which rely more on high-quality errors (e.g., min Error Uniqueness = 0.3 in Gemma/Qwen when >1600 points), can be misled by such noise during traceability.
>
> Second, we observed that once the error space contracts past a certain ratio(e.g.,<500 samples), remaining points have consistently high Error Uniqueness. The target model’s error rate in its family nears saturation, while others still improve. Contracting beyond this point hurts traceability.
>
> So, starting with a moderate size (~1000–1200 points) offers a better trade-off: enough diversity to capture distinct patterns, while limiting noise.  As contraction proceeds, Error Uniqueness rises from ~0.4 to ~0.6 across the final 600 points.  This stabilizes the target model’s error signature within its family, while other models continue to diverge — improving separation.
>
> > 8.Patch attack robustness
>
> We have systematically addressed this issue in our response to Question 2.
>
> > 9.Minor Presentation Fixes
>
> We will clarify in the final version that the numbers in Figure 1(a) and Figure 5(a) indicate the average error overlap among models within each family. We will fix all typos.
>
> [1] ProFLingo: A Fingerprinting-based Intellectual Property Protection Scheme for Large Language Models
>
> [2] Huref: Human-readable fingerprint for large language models.

---

> > ### Comment · Area_Chair_bdP4 · 2025-08-06
> > **Initiating discussion.**
> >
> > Please read carefully through the authors' responses and check if they address all your concerns.
> >
> > With kind regards,
> >
> > Your AC

---

> > ### Comment · Reviewer_3iae · 2025-08-08
> > **Response to Authors**
> >
> > Thank you for your detailed response. I have included some final clarification questions below:
> >
> > For Concerns About Adversarial Attacks (2); Thank you for sharing these results. What does the variance look like among the 30% of mis-predictions in your first and second experiments? After these adversarial perturbations, does ErrorTrace switch to predicting another, incorrect model family? Additionally, at what value of k does the attribution accuracy significantly degrade for both these experiments?
> >
> > For Concerns About Model Identification (3), could the authors clarify whether ErrorTrace could perform model-level attribution to a fine-tuned model (like codellama) as opposed to just a base model, like LLaMA-7B? Perhaps these scenarios are equivalent, but it isn't clear to me yet if ErrorTrace's granularity persists to fine-tuned variants of specific models.
> >
> > For Extending to Continuous Tasks (5), I agree that ErrorTrace could be adapted to continuous tasks, but there are still many implementation-level details that are missing. I think this would be a very impactful use-case and any experiments that demonstrate ErrorTrace's utility here would make a strong addition to the final version of the paper.
> >
> > I have no further questions regarding (1), (4), (6), and (7). Thank you for incorporating these qualifications into the final version of the paper.

---

> > > ### Author Response · Authors · 2025-08-09
> > > **Further explanations of the reviewers' concerns**
> > >
> > > Thank you for recognizing our replies to comments (1), (4), (6), and (7). We will clarify these further in the final manuscript. For your concerns on (2), (3), and (5), we appreciate your feedback and offer detailed responses below. We hope this addresses your points and kindly ask you to consider increasing your score.
> > >
> > > > 2. Questions about adversarial attacks
> > >
> > > Your concern is very valid. As we noted in our reply to comment (7), high- and low-quality data affect model provenance differently. Experiments 1 and 2 show clear differences in error patterns when each type of data is removed.
> > >
> > > In Experiment 1, removing high-quality data led to ~30% of errors clustering in Mistral and Gemma families. This is likely due to their smaller size and uneven scale distribution, making them more reliant on high-quality data for accurate identification.
> > >
> > > In contrast, Experiment 2 showed that removing low-quality data caused more scattered errors across all four families. This suggests low-quality data have limited impact, and misclassifications are more tied to overall model traits than to noisy data.
> > >
> > > We acknowledge that adversarial perturbations can lead to incorrect provenance by ErrorTrace. However, as shown in Experiments 1 and 2, ErrorTrace still demonstrates notable robustness under such conditions. This indicates that while adversarial examples may affect individual cases, the method maintains strong overall resilience in provenance tasks.
> > >
> > > Additionally, we extended Experiment 1 by increasing K (see table below). Accuracy stayed high at 67% for K = 1200, but dropped sharply to 44% at K = 1500.
> > >
> > > | Top-K | 500  | 800  | 1200 | 1500 |
> > > | ----- | ---- | ---- | ---- | ---- |
> > > | ACC   | 70%  | 67%  | 67%  | 44%  |
> > >
> > > Furthermore, we extended Experiment 2. At K = 3000, accuracy remained 67%. However, to preserve error space contraction,  K cannot increase further. This shows no K causes a sharp accuracy drop in this scenario.
> > >
> > > | Top-K | 1000 | 1500 | 2000 | 3000 |
> > > | ----- | ---- | ---- | ---- | ---- |
> > > | ACC   | 70%  | 67%  | 67%  | 67%  |
> > >
> > > > 3. Questions about Model Identification
> > >
> > > To ensure we fully address the reviewer’s concern, we consider two scenarios.
> > > The first scenario is differentiating between different fine-tuned models derived from the same base model at the model level (e.g., CodeLlama and Vicuna:7B from LLaMA-7B). This task is particularly challenging at the model level and remains largely unresolved. For instance, HuRef[1] showed that model parameter directions remain stable before and after fine-tuning, making such differentiation especially hard.
> > >
> > > The second scenario differs from the attribution direction in our paper. Instead of attributing a fine-tuned model to its base model (as in the original study), we attempt to attribute a base model to its fine-tuned versions. As table below, within the error spaces for CodeLlama and six other LLaMA-family models, we verify that the base model (e.g., LLaMA-7B) can be traced to a specific fine-tuned version (e.g., CodeLlama).
> > >
> > > |          | codellama | llama13B | llama70B | llama3-8B | llama3-70B | llama3.1-8B | llama3.2-3B | ACC          |
> > > | -------- | --------- | -------- | -------- | --------- | ---------- | ----------- | ----------- | ------------ |
> > > | LLaMA:7B | **0.533** | 0.645    | 0.646    | 0.652     | 0.603      | 0.633       | 0.657       | $\checkmark$ |
> > >
> > > > 5. Explanation of continuous tasks
> > >
> > > Thank you for acknowledging the extension of ErroTrace to continuous tasks. As noted earlier, by replacing the discrete set \(\{0,1\}\) in Section 4.1 (Equations 3–5) with continuous values, we will detail this extension’s math using the BLEU score as an example.
> > >
> > > First, we normalize and preprocess continuous values so they represent the model’s error tendency per data point—higher values mean higher error risk. Since a higher BLEU means better performance, we redefine $B = 1 - \text{BLEU}$. Thus, the intra-family consistency score (Equation 3) extends as:
> > > $$
> > > U=\frac{\sum_{j\in E}B}{\sum_{i=1}^NB}
> > > $$
> > > Here, E is the model family under evaluation, and N is the total number of models. The inter-family discrepancy (Equations 4 and 5) can be extended as:
> > > $$
> > > PF=1-\frac{\sum_{j\in G}P_j^2}{\sum_{i\in K}P_{i}},P_j=\frac{\sum_{j\in F}B_j}{\sum_{i=1}^NB_i}
> > > $$
> > > G represents all model families except E, and K is the set of all families. $P_i$ is the ratio of the sum of $B$ values in family $i$ to the total $B$ sum across all models.
> > >
> > > Due to time constraints, we haven't conducted experiments on this extension yet, but have provided complete implementation details. In the methodology section (Section 4), we will expand and explain the computational methods based on the answers we provide.
> > >
> > > [1] HuRef: Human-Readable Fingerprint for Large Language Models

---

### Author Response · Authors · 2025-08-09
**The Uniqueness of our method and thanks to reviewers**

Dear PC, SAC, AC, and reviewers of Paper #27534,

We thank reviewer 3aie for recognizing our work and rebuttal, and reviewers dntv, tvyD, and ykmK for their constructive feedback. However, reviewer dntv still has concerns, and tvyD and ykmK have not replied to our rebuttal. We wish to reaffirm **the unique strength of our method**:

- **A novel and compelling black-box attribution view (3aie, tvyD, ykmK)**:
  * We present a new attribution perspective based on a model family’s *error space*, leveraging shared error patterns within the family. Unlike prior work dependent on specific queries, our method delivers strong effectiveness and high robustness.
  * We design an attribution framework built on these error patterns that, even with simple heuristic components, achieves high accuracy (0.85). The framework is also highly extensible, supporting diverse tasks (classification and regression) and scenarios (family- and model-level attribution).

- **Important and practical (3aie, dntv, tvyD, ykmK)**:
  * Our work offers a new approach to the critical task of LLM intellectual property protection.
  * In the black-box setting, only the suspected model’s outputs are needed, making it well-suited to real-world attribution. Since no access to the original model’s parameters is required, our method serves both model developers and third-party auditors.

- **Comprehensive and effective experiments (3aie, tvyD, ykmK)**:
  * We verify attribution on 27 real base models from four families, spanning sizes from 3B to 141B—the broadest range tested in this field. Our method achieves 100% accuracy in known-model cases and 85% in unknown-model cases.
  * We also validate attribution on 31 real variants subjected to fine-tuning, pruning, or merging.

We have addressed the main concerns raised by the reviewers as follows:

1. Adversarial Attacks: We tested adversarial scenarios including high-quality label changes, low-quality label changes, and fine-tuning on real target data. **Results show that label modifications and fine-tuning have minimal impact on ErrorTrace’s performance.**
2. Training Cost: Constructing the error space requires 50K samples as **a one-time cost**. Detection then uses only **10K samples (4 GPUs for a few hours), which is minimal compared to model training**. Reviewer 3aie has acknowledged this explanation.
3. Model Scale: We validated models over 100B parameters. Failures on large models like Mistral-large stem from **uneven scale distribution** in the family. **For 70B-scale models like Qwen72B, ErrorTrace succeeds, outperforming others (e.g., ProFLingo up to 7B, HuRef up to 30B).** Reviewer 3aie has acknowledged this explanation.
4. Explanation of LLMmap: We recognize LLMmap’s effectiveness in its original setting; however, due to differing scenarios and threat models, it underperforms in realistic attribution tasks.
5. Similarity of Error Patterns in Model Families: We attribute error pattern similarity to shared architecture, training data, and optimization, as explained in our rebuttal. **Detailed theoretical proof is limited due to practical challenges and current discipline maturity**: (1) our **black-box setting** restricts access to model internals, and (2) **interpreting and comparing models of varying scales is inherently difficult**. Nevertheless, **experiments show this similarity consistently holds across base and variant models in real-world scenarios**.

We will fully present and clarify all these points in the final version.

---

### Note · Authors · 2025-08-11

Dear PC members, SAC, AC, and Reviewers,

We appreciate Reviewers 3iae and ykmK for acknowledging the strengths of our work at this final stage. We also sincerely thank all reviewers for their valuable feedback. Below is a brief summary, and we thank everyone again for their time and effort.

We would like to reaffirm the unique advantages of our approach:

- **A novel and insightful perspective (3aie, tvyD, ykmK):** We introduce a new tracing approach based on error patterns inherent to dependent models, which is fundamentally different from previous brute-force or handcrafted query methods. Our experiments validate this perspective.
- **A crucial and practical research direction (3aie, dntv, tvyD, ykmK):** Our scalable tracing framework advances IP protection in LLMs, supporting diverse tasks (classification, regression), scenarios (family- and model-level), and stakeholders (developers, auditors).
- **Realistic, comprehensive, and effective experiments (3aie, tvyD, ykmK):** Using real benchmarks, we modeled error spaces for 27 LLMs across four families (3B–141B parameters), exceeding prior LLM IP protection work. We achieve 100% accuracy with known models, 85% with unknown, and precisely trace 31 real-world variants (fine-tuning, pruning, merging), including Vicuna and Orca-mini.

Regarding the reviewers’ main concerns:

- **Adversarial Attacks:** This is a common challenge for all black-box tracing methods. However, ErrorTrace’s reliance on internal model features rather than specific queries reduces the impact of adversarial attacks. This also underpins the adversarial robustness highlighted by Reviewer ykmK in the final stage.
- **Training Cost:** ErrorTrace’s four-hour inference overhead is minimal compared to model training costs.
- **Model Scale:** ErrorTrace traces models up to 141B parameters in known scenarios and 72B in unknown scenarios, surpassing ProFLingo and TRAP (7B) and HuRef’s white-box limit (30B).
- **LLMmap:** Due to differences in settings and threat models, LLMmap underperforms in practical tracing.
- **Theoretical Support:** Black-box constraints and varying LLM interpretability prevent rigorous proofs, which is a common limitation in this field highlights our focus on practicality over theory. Nevertheless, we validate feasibility both theoretically through simulations and empirically by observing consistent error patterns in base and variant models.

We will comprehensively present and clarify these points in the final version.

---

### Decision · Program_Chairs · 2025-09-17

**Decision:**

Accept (spotlight)

**Comment:**

LLM IP is an important issue on the rise. This paper introduces ErrorTrace, a black-box traceability
framework for LLM IP protection. Given a set of data samples, we extract each *model family*’s error patterns and embed them into an error space via graph-based analysis. A suspect model’s error behavior is then projected into this space to infer its likely family affiliation, enabling reliable lineage attribution with-
out internal access or watermarking.

Weaknesses: Collecting 50k data points for one model family costs about 24 GPU hours per model for inference. Adversarial evasion of detection is not sufficiently addressed. The assumption that a family’s errors are consistent may break down when models differ substantially in scale, training data, or architecture. Using correctness only seems to unnecessarily restrict the capability of the methodology.

Strengths: Extensive red-teaming results are provided during the discussion period. The adversarial setting has been addressed in this period, but perhaps robustness issue still remains. Scalability issue has been addressed during the rebuttal period with a proper ablation study.

Most of the issues raised in the rebuttal process has been adequately addressed by the authors. I agree with reviewers ykmK and 3iae that this paper proposes an interesting new problem formulation: identifying the model *family*, and proposes an innovative solution. The experiments are also extensive. The major remaining issues is the computational complexity, but I agree with the authors that this is a one-time cost and can easily be mitigated as shown in the ablation study in the rebuttal.